# Offline Contextual Bandits with High Probability Fairness Guarantees

**Blossom Metevier[1]**   **Stephen Giguere[1]**   **Sarah Brockman[1]**   **Ari Kobren[1]**
**Yuriy Brun[1]**       **Emma Brunskill[2]**       **Philip S. Thomas[1]**
[1]College of Information and Computer Sciences       [2]Computer Science Department
University of Massachusetts Amherst                    Stanford University

## Abstract

We present `RobinHood`, an offline contextual bandit algorithm designed to satisfy a broad family of fairness constraints. Our algorithm accepts multiple fairness definitions and allows users to construct their own unique fairness definitions for the problem at hand. We provide a theoretical analysis of `RobinHood`, which includes a proof that it will not return an unfair solution with probability greater than a user-specified threshold. We validate our algorithm on three applications: a user study with an automated tutoring system, a loan approval setting using the Statlog German credit data set, and a criminal recidivism problem using data released by ProPublica. To demonstrate the versatility of our approach, we use multiple well-known and custom definitions of fairness. In each setting, our algorithm is able to produce fair policies that achieve performance competitive with other offline and online contextual bandit algorithms.

## 1   Introduction

Machine learning (ML) algorithms are increasingly being used in *high-risk* decision making settings, such as financial loan approval [7], hiring [37], medical diagnostics [12], and criminal sentencing [3]. These algorithms are capable of unfair behavior, and when used to guide policy and practice, can cause significant harm. This is not merely hypothetical: ML algorithms that influence criminal sentencing and credit risk assessment have already exhibited racially-biased behavior [3; 4]. Prevention of unfair behavior by these algorithms remains an open and challenging problem [14; 20; 24]. In this paper, we address issues of unfairness in the *offline contextual bandit* setting, providing a new algorithm, designed using the recently proposed Seldonian framework [47] and called `RobinHood`, which is capable of satisfying multiple fairness definitions with high probability.

Ensuring fairness in the bandit setting is an understudied problem. While extensive research has been devoted to studying fairness in classification, recent work has shown that the decisions made by fair ML algorithms can affect the well-being of a population over time [34]. For example, criminal sentencing practices affect criminal recidivism rates, and loan approval strategies can change the amount of wealth and debt in a population. This delayed impact indicates that the feedback used for training these algorithms or defining fairness is more *evaluative* in nature, i.e., training samples quantify the (delayed) outcome of taking a particular action given a particular context. Therefore, it is important that fairness can be ensured for methods that are designed to handle evaluative feedback, such as contextual bandits. For example, instead of predicting the likelihood of violent recidivism, these methods can consider what actions to take to minimize violent recidivism.

Within the bandit setting, prior work has mostly focused on ensuring fairness in the *online* setting [24; 25; 27; 35], in which an *agent* learns the quality of different actions by interacting with the system of interest. However, for many fairness applications, e.g., medical treatment suggestion [29], the online setting is not feasible, as direct interaction might be too costly, risky, or otherwise unrealistic. Instead,

these problems can be framed in the *offline* bandit setting, where a finite amount of data is collected from the system over time and then used by the agent to construct a fair solution.

Issues of ensuring fairness in the offline contextual bandit setting are similar to those in other ML settings. For instance, contextual bandit algorithms need to manage the trade-off between performance optimization and fairness. Ideally, these algorithms should also be capable of handling a large set of user-defined fairness criteria, as no single definition of fairness is appropriate for all problems [16]. Importantly, when it is not possible to return a fair solution, i.e., when fairness criteria are in conflict, or when too little data is present, the algorithm should indicate this to the user. We allow our algorithm, RobinHood, to return *No Solution Found* (NSF) in cases like this, and show that if a fair solution *does* exist, the probability RobinHood returns NSF goes to zero as the amount of available data increases.

In summary, we present the first Seldonian algorithm for contextual bandits. Our contributions are: **1)** we provide an offline contextual bandit algorithm, called RobinHood, that allows users to mathematically specify their own notions of fairness, including combinations of fairness definitions already proposed by the ML community, and novel ones that may be unique to the application of interest; **2)** we prove that RobinHood is (quasi-)Seldonian: that it is guaranteed to satisfy the fairness constraints defined by the user with high probability, **3)** we prove that if a fair solution exists, as more data is provided to RobinHood, the probability it returns NSF goes to zero; and **4)** we evaluate RobinHood on three applications: a tutoring system in which we conduct a user study and consider multiple, unique fairness definitions, a loan approval setting in which well-known fairness definitions are applied, and criminal recidivism. Our work complements fairness literature for classification (e.g., [1; 8; 14; 50]), contextual bandits (e.g., [24; 25; 35; 14; 27; 22]), and reinforcement learning (e.g., [40; 49]), as described in Section 7.

## 2 Contextual Bandits and Offline Learning

This section defines a contextual bandit, or *agent*, which iteratively interacts with a system. At each iteration $\iota \in \{1, 2, ...\}$, the agent is given a context, represented as a random variable $X_\iota \in \mathbb{R}^\nu$ for some $\nu$. We assume that the contexts during different iterations are *independent and identically distributed* (i.i.d.) samples from some distribution, $d_X$. Let $\mathcal{A}$ be the finite set of possible actions that the agent can select. The agent's *policy*, $\pi : \mathbb{R}^\nu \times \mathcal{A} \to [0, 1]$ characterizes how the agent selects actions given the current context: $\pi(x, a) = \Pr(A_\iota = a | X_\iota = x)$. Once the agent has chosen an action, $A_\iota$, based on context $X_\iota$, it receives a stochastic real-valued reward, $R_\iota$. We assume that the conditional distribution of $R_\iota$ given $X_\iota$ and $A_\iota$ is given by $d_R$, i.e., $R_\iota \sim d_R(X_\iota, A_\iota, \cdot)$. The agent's goal is to choose actions so as to maximize the expected reward it receives.

For concreteness, we use a loan approval problem as our running example. For each loan applicant (iteration $\iota$), a single action, i.e., whether or not the applicant should be given a loan, is chosen. The resulting reward is a binary value: 1 if the loan is repaid and $-1$ otherwise. The agent's goal using this reward function is to maximize the expected number of loan repayments.

This paper focuses on enforcing fairness in the offline setting, where the agent only has access to a finite amount of *logged* data, $D$, collected from one or more different policies. Policies used to collect logged data are known as *behavior* policies. For simplicity of notation, we consider a single behavior policy, $\pi_b$. $D$ consists of the observed contexts, actions, and rewards: $D = \{(X_\iota, A_\iota, R_\iota)\}_{\iota=1}^m$, where $m$ is the number of iterations for which $D$ was collected, and where $A_\iota \sim \pi_b(X_\iota, \cdot)$.

The goal in the offline setting is to find a policy, $\pi'$, which maximizes $r(\pi') := \mathbf{E}[R_\iota | A_\iota \sim \pi'(X_\iota, \cdot)]$ using samples from $D$ only. Algorithms that solve this problem are called offline (or batch) contextual bandit algorithms. At the core of many offline contextual bandit algorithms is an *off-policy estimator*, $\hat{r}$, which takes as inputs $D$ and a policy to be evaluated, $\pi_e$, in order to produce an estimate, $\hat{r}(\pi_e, D)$, of $r(\pi_e)$. We call $\hat{r}(\pi_e, D)$ the off-policy reward of $\pi_e$.

## 3 Problem Statement

Following the Seldonian framework for designing ML algorithms [47], our goal is to develop a fair offline contextual bandit algorithm that satisfies three conditions: **1)** the algorithm accepts multiple user-defined fairness definitions, **2)** the algorithm guarantees that the probability it returns a policy that violates each definition of fairness is bounded by a user-specified constant, and **3)** if a fair solution

exists, the probability that it returns a fair solution (other than NSF) converges to one as the amount of training data goes to infinity. The first condition is crucial because no single definition of fairness is appropriate for all problems [16]. The second condition is equally important because it allows the user to specify the necessary confidence level(s) for the application at hand. However, if an algorithm only satisfies the first two conditions, this does not mean it is qualitatively helpful. For example, we can construct an algorithm that always returns NSF instead of an actual solution—this technically satisfies **1)** and **2)**, but is effectively useless. Ideally, if a fair solutions exists, a fair algorithm should be able to (given enough data) eventually find and return it. We call this property *consistency* and show in Section 5 that RobinHood is consistent.

Since condition **1)** allows users to specify their own notions of fairness, the set of fair policies is not known *a priori*. Therefore, the algorithm must reason about the fairness of policies using the available data. We consider a policy to be either fair or unfair with respect to condition **2)**. For example, we may say that a lending policy is fair if for all pairs of applicants who are identical in every way except race, the policy takes the same action, i.e., approving or denying the loan for both. This criterion is known as causal discrimination (with respect to race) [17; 30]. Then, a policy that adheres to this definition is fair, and a policy that violates this definition is unfair. The algorithm that produces a policy from data must ensure that, with high probability, it will produce a policy that is fair. Note that the setting in which each *action* is fair or unfair can be captured by this approach by defining a policy to be fair if and only if the probability it produces unfair actions is bounded by a small constant.

Formally, assume that policies are parameterized by a vector $\theta \in \mathbb{R}^l$, so that $\pi(X_\iota, \cdot, \theta)$ is the conditional distribution over action $A_\iota$ given $X_\iota$ for all $\theta \in \mathbb{R}^l$. Let $\mathcal{D}$ be the set of all possible logged data sets and $D$ be the logged data (a random variable, and the only source of randomness in the subsequent equations). Let $a : \mathcal{D} \to \mathbb{R}^l$ be an offline contextual bandit algorithm, which takes as input logged data and produces as output the parameters for a new policy.

Let $g_i : \mathbb{R}^l \to \mathbb{R}$ be a user-supplied function, called a *constraint objective*, that measures the fairness of a policy. We adopt the convention that if $g_i(\theta) \leq 0$ then the policy with parameters $\theta$ is fair, and if $g_i(\theta) > 0$ then the policy with parameters $\theta$ is unfair. Our goal is to create a contextual bandit algorithm that enforces $k$ behavioral constraints, where the $i^{\text{th}}$ constraint has the form:

$$\Pr\Big(g_i(a(D)) \leq 0\Big) \geq 1 - \delta_i, \tag{1}$$

where $\delta_i \in [0, 1]$ is the required confidence level for the $i^{\text{th}}$ constraint. Together, the constraint objective $g_i$ and the confidence level $\delta_i$ constitute a *behavioral constraint*. Any algorithm that satisfies (1) is called *Seldonian*, or *quasi-Seldonian* if it relies on reasonable but false assumptions, such as appeals to the central limit theorem [47].

Some constraints might be impossible to enforce if, for example, the user provides conflicting constraints [28], or if $\delta_i$ cannot be established given the amount of data available. Algorithms that provide high-probability guarantees are often very conservative [26]; if only a small amount of data is available, it may be impossible for the algorithm to produce a policy that satisfies all behavioral constraints with sufficiently high confidence. In such cases, RobinHood returns NSF to indicate that it is unable to find a fair solution. When this occurs, the user has control over what to do next. For some domains, deploying a known fair policy may be appropriate; for others, it might be more appropriate to issue a warning and deploy no policy. We define $g_i(\text{NSF}) = 0$, so that NSF is always fair.

Notice that the creation of an algorithm that satisfies each of the three desired conditions is difficult. Condition **1)** is difficult to enforce because the user must be provided with an interface that allows them to specify their desired definition of fairness without requiring the user to know the underlying data distribution. Condition **2)** is difficult to achieve because of the problem of multiple comparisons— testing $b$ solutions to see if they are fair equates to performing $b$ hypothesis tests, which necessitates measures for avoiding the problems associated with running multiple hypothesis tests using a single data set. Condition **3)** is particularly difficult to achieve in conjunction with the second condition— the algorithm must carefully trade-off the maximizing expected reward with predictions about the outcomes of future hypothesis tests when picking the candidate solution that it considers returning.

## 4  RobinHood Algorithm

This section presents RobinHood, our fair bandit algorithm. At a high level, RobinHood allows users to specify their own notions of (un)fairness based on statistics related to the performance of a

potential solution. It then uses concentration inequalities [36] to calculate high-probability bounds on these statistics. If these bounds satisfy the user's fairness criteria, then the solution is returned.

**Constructing Constraint Objectives.** Users can specify their desired fairness definitions with constraint objectives, $\{g_i\}_{i=1}^k$, that accept a parameterized policy $\theta$ as input and produce a real-valued measurement of fairness. For simplicity of notation, we remove the subscript $i$ and discuss the construction of an arbitrary constraint objective, $g$. In our loan approval example, we might define $g(\theta) = \text{CDR}(\theta) - \epsilon$, where $\text{CDR}(\theta)$ indicates the causal discrimination rate of $\theta$. However, computing $g$ for this example requires knowledge of the underlying data distribution, which is typically unknown. In practice, each $g$ might depend on distributions that are unavailable, so it is unreasonable to assume that the user can compute $g(\theta)$ for an arbitrary $g$.

Instead of explicitly requiring $g(\theta)$, we could instead assume that the user is able to provide unbiased estimators for $g$. However, this is also limiting because it may be difficult to obtain unbiased estimators for certain constraint objectives, e.g., unbiased estimators of the standard deviation of a random variable can be challenging (or impossible) to construct. Importantly, our algorithm does *not* explicitly require an unbiased estimate of $g(\theta)$. Instead, it computes high-probability upper bounds for $g(\theta)$. Even if the random variable of interest does not permit unbiased estimators, if it is a function of random variables for which unbiased estimators exist, then valid upper bounds can be computed.

With this in mind, we propose a general interface in which the user can define $g$ by combining $d$ real-valued *base variables*, $\{z_j\}_{j=1}^d$, using addition, subtraction, division, multiplication, absolute value, maximum, inverse, and negation operators. Base variables may also be multiplied and added to constants. Instead of specifying the base variable $z_j(\theta)$ explicitly, we assume the user is able to provide an unbiased estimator, $\hat{z}_j$, of each base variable: $z_j(\theta) := \mathbf{E}[\text{Average}(\hat{z}_j(\theta, D))]$. That is, each function $\hat{z}_j$ takes a parameterized policy $\theta$ and a data set $D$ and returns an arbitrary number of i.i.d. outputs, $\hat{z}_j(\theta, D)$. In the definition of $z_j$, the average of the outputs is taken so $z_j(\theta)$ is a scalar.

A base variable estimator $\hat{z}_j$ for our loan approval example could be defined as an integer indicating whether or not causal discrimination is satisfied for applicable data points in $D$. To do this, $\hat{z}(\theta, D)$ should produce 1 if for points $h$ and $f$ that differ only by race, $\theta$ chooses the same action, and 0 otherwise. Defining $\hat{z}$ in this way gives us an unbiased estimate of the CDR. We could then define $g(\theta) = z(\theta) - \epsilon$, requiring the CDR to be within some value $\epsilon$, with probability at least $1 - \delta$.

There may be some base variables that the user wants to use when defining fairness that do not have unbiased estimators, e.g., standard deviation. To handle these cases, we also allow the user to use base variables for which they can provide high-probability upper and lower bounds on $z(\theta)$ given any $\theta$ and data $D$. As an example, in Appendix G we show how the user can define a base variable to be the largest possible expected reward for any policy with parameters $\theta$ in some set $\Theta$, i.e., $\max_{\theta \in \Theta} r(\theta)$.

In summary, users can define constraint objectives that capture their desired definitions of fairness. Constraint objectives are mathematical expressions containing operators (including summation, division, and absolute value) and base variables (any variable, including constants, for which high-confidence upper and lower bounds can be computed). In Section 6, we construct constraint objectives for different fairness definitions and find solutions that are fair with respect to these definitions. In Appendix A, we provide more examples of how to construct constraint objectives for other common fairness definitions used in the ML community.

**Pseudocode.** RobinHood (Algorithm 1) has three steps. In the first step, it partitions the training data into the candidate selection set $D_c$ and the safety set $D_s$. In the second step, called *candidate selection*, RobinHood uses $D_c$ to construct a candidate policy, $\theta_c \in \Theta$, that is likely to satisfy the third step, called the *safety test*, which ensures that the behavioral constraints hold. Algorithm 1 presents the RobinHood pseudocode. Because the candidate selection step is constructed with the safety step in mind, we first discuss the safety test, followed by candidate selection. Finally, we summarize the helper methods used during the candidate selection and safety test steps.

The safety test (lines 3–4) applies concentration inequalities using $D_s$ to produce a high-probability upper bound, $U_i := U_i(\theta_c, D_s)$, on $g_i(\theta_c)$, the value of the candidate solution found in the candidate selection step. More concretely, $U_i$ satisfies $\Pr(g_i(\theta_c) \le U_i) \ge 1 - \delta_i$. If $U_i \le 0$, then $\theta_c$ passes the safety check and is returned. Otherwise, RobinHood returns NSF.

The goal of the candidate selection step (line 2) is to find a solution, $\theta_c$, that maximizes expected reward and is likely to pass the safety test. Specifically, $\theta_c$ is found by maximizing the output of

---

**Algorithm 1** RobinHood $(D, \Delta = \{\delta_i\}_{i=1}^k, \hat{Z}\{\hat{z}_j^i\}_{j=1}^d, \mathcal{E} = \{E_i\}_{i=1}^k)$

---

1: $D_c, D_s = \texttt{partition}(D)$
2: $\theta_c = \arg\max_{\theta \in \Theta} \texttt{CandidateUtility}(\theta, D_c, \Delta, \hat{Z}, \mathcal{E})$ ▷ Candidate Selection
3: $[U_1, ..., U_k] = \texttt{ComputeUpperBounds}(\theta_c, D_s, \Delta, \mathcal{E}, \texttt{inflateBounds=False})$ ▷ Safety Test
4: **if** $U_i \leq 0$ for all $i \in \{1, ..., k\}$ **then return** $\theta_c$ **else return** NSF

---

---

**Algorithm 2** CandidateUtility$(\theta, D_c, \Delta, \hat{Z}, \mathcal{E})$

---

1: $[\hat{U}_1, ..., \hat{U}_k] = \texttt{ComputeUpperBounds}(\theta, D_c, \Delta, \hat{Z}, \mathcal{E}, \texttt{inflateBounds=True})$
2: $r_{\min} = \min_{\theta' \in \Theta} \hat{r}(\theta', D_c)$

3: **if** $\hat{U}_i \leq -\xi$ for all $i \in \{1, ..., k\}$ **then return** $\hat{r}(\theta, D_c)$ **else return** $r_{\min} - \sum_{i=1}^{k} \max\{0, \hat{U}_i\}$

---

Algorithm 2, which uses $D_c$ to compute an estimate, $\hat{U}_i$, of $U_i$. The same concentration inequality used to compute $U_i$ in the safety test is also used to compute $\hat{U}_i$, e.g., Hoeffding's inequality is used to compute $U_i$ and $\hat{U}_i$. Multiple comparisons are performed on a single data set ($D_c$) during the search for a solution. This leads to the candidate selection step over-estimating its confidence that the solution it picks will pass the safety test. In order to remedy this issue, we inflate the width of the confidence intervals used to compute $\hat{U}_i$ (this is indicated by the Boolean $\texttt{inflateBounds}$ in the pseudocode). Another distinction between the candidate selection step and the safety test is that, in Algorithm 2, we check whether $\hat{U}_i \leq -\xi$ for some small constant $\xi$ instead of 0. This technical assumption is required to ensure the consistency of RobinHood, and is discussed in Appendix E.

We define the input $\mathcal{E} = \{E_i\}_{i=1}^k$ in the pseudocode to be analytic expressions representing the constraint objectives $\{g_i\}_{i=1}^k$. For example, if we had the constraint objective $g(\theta) = z_1(\theta) \times z_2(\theta) + z_3(\theta) - \epsilon$, then $E = E_1 \times E_2 + E_3 - \epsilon$, where $E_1 = z_1(\theta)$, $E_2 = z_2(\theta)$, and $E_3 = z_3(\theta)$. Each expression $E_i$ is used in $\texttt{ComputeUpperBounds}$ (Algorithm 3) and related helper functions, the pseudocode for which is located in Appendix B. At a high level, these helper functions recurse through sub-expressions of each $E_i$ until a base variable is encountered. Once this occurs, real-valued upper and lower $(1 - \delta_i)$-confidence bounds on the base variable's estimators are computed and subsequently propagated through $E_i$. Using our example from earlier, bounds for $E_{1,2} = E_1 \times E_2$ would first be computed, followed by bounds for $E_{1,2} + E_3$.

## 5 Theoretical Analyses

This section proves that given reasonable assumptions about the constraint objectives, $\{g_i(\theta)_{i=1}^k\}$, and their sample estimates, $\{\text{Average}(\hat{g}_i(\theta, D))\}_{i=1}^k$, **1)** RobinHood is guaranteed to satisfy the behavioral constraints and **2)** RobinHood is consistent.

To prove that RobinHood is Seldonian [47]: it is guaranteed to satisfy the behavioral constraints, i.e., that RobinHood returns a fair solution with high probability, we show that the safety test only returns a solution if the behavioral constraints are guaranteed to be satisfied. This follows from the use of concentration inequalities and transformations to convert bounds on the base variables, $z_j(\theta_c)$, into a high-confidence upper bound, $U_i$, on $g_i(\theta_c)$. We therefore begin by showing (in Appendix C) that the upper bounds computed by the helper functions in Appendix B satisfy $\Pr(g_i(\theta) > U_i) \leq \delta_i$ for all constraint objectives. Next, we show that the behavioral constraints are satisfied by RobinHood. We denote RobinHood as $a(D)$, a batch contextual bandit algorithm dependent on data $D$.

**Theorem 1** (Fairness Guarantee). *Let $\{g_i\}_{i=0}^n$ be a sequence of behavioral constraints, where $g_i : \Theta \to \mathbb{R}$, and let $\{\delta_i\}_{i=0}^n$ be a corresponding sequence of confidence thresholds, where $\delta_i \in [0, 1]$. Then, for all $i$, $\Pr(g_i(a(D)) > 0) \leq \delta_i$.* **Proof**. See Appendix D.

We define an algorithm to be consistent if, when a fair solution exists, the probability that the algorithm returns a solution other than NSF converges to 1 as the amount of training data goes to infinity. We state this more formally in Theorem 2.

**Theorem 2** (Consistency). *If Assumptions 1 through 5 (specified in Appendix E) hold, then* $\lim_{n \to \infty} \Pr(a(D) \neq \mathtt{NSF},\ g(a(D)) \leq 0) = 1$. **Proof.** See Appendix E.

In order to prove Theorem 2, we first define the set $\bar{\Theta}$, which contains all unfair solutions. At a high level, we show that the probability that $\theta_c$ satisfies $\theta_c \notin \bar{\Theta}$ converges to one as $n \to \infty$. To establish this, we **1)** establish the convergence of the confidence intervals for the base variables, **2)** establish the convergence of the candidate objective for all solutions, and **3)** establish the convergence of the probability that $\theta_c \notin \bar{\Theta}$. Once we have this property about $\theta_c$, we establish that the probability of the safety test returning NSF converges to zero as $n \to \infty$.

In order to build up the properties discussed above, we make a few mild assumptions, which we summarize here. To establish **1)**, we assume that the confidence intervals on the base variables converge almost surely to the true base variable values for all solutions. Hoeffding's inequality and Student's $t$-test are examples of concentration inequalities that provide this property. We also assume that the user-provided analytic expressions, $\mathcal{E}$, are continuous functions of the base variables. With the exception of division, all operators discussed in Section 4 satisfy this assumption. In fact, this assumption is still satisfied for division when positive base variables are used in the denominator. To establish **2)** and **3)**, we assume that the sample off-policy estimator, $\hat{r}$, converges almost surely to $r$, the actual expected reward. This is satisfied by most off-policy estimators [46]. We also make particularly weak smoothness assumptions about the output of Algorithm 2, which only requires the output of Algorithm 2 to be smooth across a countable partition of $\Theta$. Lastly, we assume that at least one fair solution exists and that this solution is not on the fair-unfair boundary.

Note that consistency does not provide bounds on the amount of data necessary for the return of a fair solution. Although a high-probability bound on how much data our algorithm requires to return a solution other than NSF would provide theoretical insights into the behavior of our algorithm, our focus is on ensuring that our algorithm can return solutions other than NSF using practical amounts of data on real problems. Hence, in the following section we conduct experiments with real data (including data that we collected from a user study).

# 6   Empirical Evaluation

We apply RobinHood to three real-world applications: tutoring systems, loan approval, and criminal recidivism. Our evaluation focuses on the three research questions. **1)** When do solutions returned by RobinHood obtain performance comparable to those returned by state-of-the-art methods? **2)** How often does RobinHood return an unfair solution, as compared to state-of-the-art methods? **3)** In practice, how often does RobinHood return a solution besides NSF?

## 6.1   Experimental Methodology and Application Domains

To the best of our knowledge, no other fair contextual bandit algorithms have been proposed in the offline setting. We therefore compare to two standard offline methods: POEM [44] and Offset Tree [6]. In Appendix F, we also compare to Rawlsian Fairness, a fair *online* contextual bandit framework [23]. Existing offline bandit algorithms are not designed to adhere to multiple, user-defined fairness constraints. One seemingly straightforward fix to this is to create an algorithm that uses all of the data, $D$, to search for a solution, $\theta$, that maximizes the expected reward (using a standard approach), but with the additional constraint that an estimate, $\hat{g}(\theta, D)$, of how unfair $\theta$ is, is at most zero. That is, this method enforces the constraint $\hat{g}(\theta, D) \leq 0$ without concern for how well this constraint generalizes to future data. We construct this method, called NaïveFairBandit, as a baseline for comparison. In RobinHood and NaïveFairBandit, we use Student's $t$-test to calculate upper and lower confidence bounds on base variables for a particular $\theta$.

Note that Algorithm 1 relies on the optimization algorithm (and implicitly, the feasible set) chosen by the user to find candidate solutions. If the user chooses an optimization algorithm incapable of finding fair solutions, e.g., they choose a gradient method when the fairness constraints defined make it difficult or impossible for it to find a solution, then RobinHood will return NSF. We chose CMA-ES [19] as our optimization method. Further implementation details, including pseudocode for NaïveFairBandit, can be found in Appendix F.

**Tutoring Systems.** Our first set of experiments is motivated by *intelligent tutoring systems* (ITSs), which aim to teach a specific topic by providing personalized and interactive instruction based on a student's skill level [38]. Such adaptations could have unwanted consequences, including inequity with respect to different student populations [13]. We conduct our experiments in the *multi-armed bandit* setting—a special case of the contextual bandit setting in which context is the same for every iteration. To support these experiments, we collected user data from the crowdsourcing marketplace Amazon Mechanical Turk. In our data collection, users were presented with one of two different versions of a tutorial followed by a ten-question assessment. Data including gender, assessment score, and tutorial type was collected during the study. Let $R_\iota$ be the assessment score achieved by user $\iota$ and $S_\iota \in \{\mathtt{f}, \mathtt{m}\}$ represent the gender of user $\iota$. (Due to lack of data for users identifying their gender as "other," we restricted our analysis to male- and female-identifying users.) $D$ was collected using a uniform-random behavior policy. Further details are provided in Appendix F.

To demonstrate the ability of our algorithm to satisfy multiple and novel fairness criteria, we develop two behavioral constraints for these experiments: $g_f(\theta) := |F|^{-1} \sum_{\iota=0}^{|D|} R_\iota \mathbb{I}(\mathtt{f}) - \mathbf{E}[R|\mathtt{f}] - \epsilon_f$ and $g_m(\theta) := |M|^{-1} \sum_{\iota=0}^{|D|} R_\iota \mathbb{I}(\mathtt{m}) - \mathbf{E}[R|\mathtt{m}] - \epsilon_m$, where $|F|$ and $|M|$ denote the respective number of female- and male-identifying users, $\mathbb{I}(x)$ is an indicator function for the event $X_\iota = x$, where $x \in \{\mathtt{f}, \mathtt{m}\}$, and $\mathbf{E}[R|x]$ is the expected reward conditioned on the event $X_\iota = x$ for $x$ previously defined. In words, for the constraint $g_f$ to be less than 0, the expected reward for females may only be smaller than the empirical average for females in the collected data by at most $\epsilon_f$. The second constraint $g_m$ is similarly defined with respect to males. Note that different values of $\epsilon_f$ and $\epsilon_m$ can allow for improvement to female performance and decreased male performance and vice versa. Defining the constraints in this way may be beneficial if the user is aware that bias towards a certain group exists, and then hypothesizes that performance towards this group may need to decrease to improve performance of a minority group, as is the case in this data set. We also highlight that a fair policy for this experiment is one such that $g_f(\theta) \leq 0$ and $g_m(\theta) \leq 0$.

In our first experiment, which we call the *similar proportions* experiment, males and females are roughly equally represented in $D$: $|F| \approx |M|$. In our second ITS experiment, which we call the *skewed proportions* experiment, we simulate the scenario in which females are under-represented in the data set (and elaborate on the process for doing this in Appendix F). We perform this experiment because **1)** biased data collection is a common problem [9] and **2)** methods designed to maximize reward may do so at the expense of under-represented groups.

In the skewed proportions experiment we introduce a purposefully biased tutorial that responds differently to users based on their gender identification in order to experiment with a setting where unfairness is likely to occur. This tutorial provides information to male-identifying users in an intuitive, straightforward way but gives female-identifying users incorrect information, resulting in high assessment scores for males and near-zero assessment scores for females. The average total score for the biased tutorial in $D$ is higher than that of the other tutorials—because of this, methods that optimize for performance without regard for $g_m$ and $g_f$ will often choose the biased tutorial for deployment. In practice, a similar situation could occur in an adaptive learning system, where tutorials are uploaded by different sources. The introduction of a bug might compel a fairness-unaware algorithm to choose an unfair tutorial.

**Loan Approval.** Our next experiments are inspired by decision support systems for loan approval. In this setting, a policy uses a set of features, which describe an applicant, to determine whether the applicant should be approved for a loan. We use the Statlog German Credit data set, which includes a collection of loan applicants, each one described by 20 features, and labels corresponding to whether or not each applicant was assessed to have good financial credit [32]. A policy earns reward 1 if it approves an applicant with good credit or denies an applicant with bad credit (the credit label of each applicant is unobserved by the policy); otherwise the agent receives a reward of $-1$. We conduct two experiments that focus on ensuring fairness with respect to sex (using the `Personal Status and Sex` feature of each applicant in the data set to determine sex). Specifically, we enforce *disparate impact* [51] in one and *statistical parity* [14] in the other. In Appendix F we define statistical parity for this domain and discuss experimental results.

Disparate impact is defined in terms of the *relative magnitudes* of positive outcome rates. Let $\mathtt{f}$ and $\mathtt{m}$ correspond to females and males and let $A = 1$ if the corresponding applicant was granted a loan and $A = 0$ otherwise. Disparate impact can then be written as: $g(\theta) :=$

$\max \{ \mathbf{E}[A|\mathtt{m}]/\mathbf{E}[A|\mathtt{f}], \mathbf{E}[A|\mathtt{f}]/\mathbf{E}[A|\mathtt{m}] \} - (1+\epsilon)$. To satisfy this, neither males nor females may enjoy a positive outcome rate that is more than $100\epsilon\%$ larger than that of the other sex.

**Criminal Recidivism.** This experiment uses recidivism data released by ProPublica as part of their investigation into the racial bias of deployed classification algorithms [3]. Each record in the data set includes a label indicating if the person would later re-offend (decile score), and six predictive features, including juvenile felony count, age, and sex. In this problem, the agent is tasked with producing the decile score given a feature vector of information about a person (the decile score label of each applicant is unobserved by the policy). The reward provided to the bandit is 1 if recidivism occurs and 0 otherwise. We apply *approximate* statistical parity here, where features of interest are race (Caucasian and African American). A policy exhibits statistical parity if the probability with which it assigns a beneficial outcome to individuals belonging to protected and unprotected classes is equal: $g(\theta) \coloneqq |\Pr(A = 1|\text{Caucasian}) - \Pr(A = 1|\text{African American})| - \epsilon$.

## 6.2 Results and Discussion

Figure 1 shows our experimental results over varying training set sizes. The leftmost plots in each row show the off-policy reward of solutions returned by each algorithm, and the middle plots show how often solutions are returned by each algorithm. The solution rate for each baseline is 100% because `RobinHood` is the only algorithm able to return `NSF`. The purpose of plotting the solution rate is to determine how much data is required for solutions other than `NSF` to be returned by our algorithm. The rightmost plots show the probability that an algorithm violated the fairness constraints. The dashed line in these plots denotes the maximum failure rate allowed by the behavioral constraints ($\delta = 5\%$ in our experiments).

In all of our experiments, unless a certain amount of data is provided to `NaïveFairBandit`, it returns unfair solutions at an unacceptable rate. This seems workable at first glance—one could argue that so long as enough data is given to `NaïveFairBandit`, it will not violate the behavioral constraints. In practice, however, it is not known in advance how much data is needed to obtain a fair solution. `NaïveFairBandit`'s failure rate varies considerably in each experiment, and it is unclear how to determine the amount of data necessary for `NaïveFairBandit`'s failure rate to remain under $\delta$. In essence, `RobinHood` is a variant of `NaïveFairBandit` that includes a mechanism for determining when there is sufficient data to trust the conclusions drawn from the available data.

In some of our experiments, the failure rates of the fairness-unaware baselines (Offset Tree and POEM) approach $\delta$ as more data is provided. To explain this behavior, note that when reward maximization and fairness are *non*conflicting, there may exist fair high-performing solutions. In the case that *only* high-performing solutions meet the fairness criteria, the failure rate of these algorithms should decrease as more data is provided. Importantly, while these baselines might be fair in some cases, unlike `RobinHood`, these approaches do not come with fairness guarantees.

In the similar proportions experiment, fairness and performance optimization are nonconflicting. In this case, `RobinHood` performs similarly to the state-of-the-art—it is able to find and return solutions whose off-policy reward is comparable to the baselines. The same pattern can be seen in the loan approval and criminal recidivism experiments. In these applications, when high-performing fair solutions exist, `RobinHood` is able to find and return them. In the skewed proportions experiment, the biased tutorial maximizes overall performance but violates the constraint objectives. As expected, POEM and Offset Tree frequently choose to deploy this tutorial regardless of the increase in training data, while `RobinHood` frequently chooses to deploy a tutorial whose off-policy reward is high (with respect to the behavioral constraints). In summary, in each of our experiments, `RobinHood` is able to return fair solutions with high probability given a reasonable amount of data.

## 7 Related Work

Significant research effort has focused on fairness-aware ML, particularly classification algorithms [1; 8; 14; 50], measuring fairness in systems [17], and defining varied notions of fairness [33; 17; 14]. Our work complements these efforts but focuses on the contextual bandit setting. This section describes work related to our setting, beginning with online bandits.

Recall (from Section 2) that in the standard online bandit setting, an agent's goal is to maximize expected reward, $\rho(a) = \mathbf{E}[R_\iota|A_\iota = a]$, as it interacts with a system. Over time, estimates of $\rho$ are

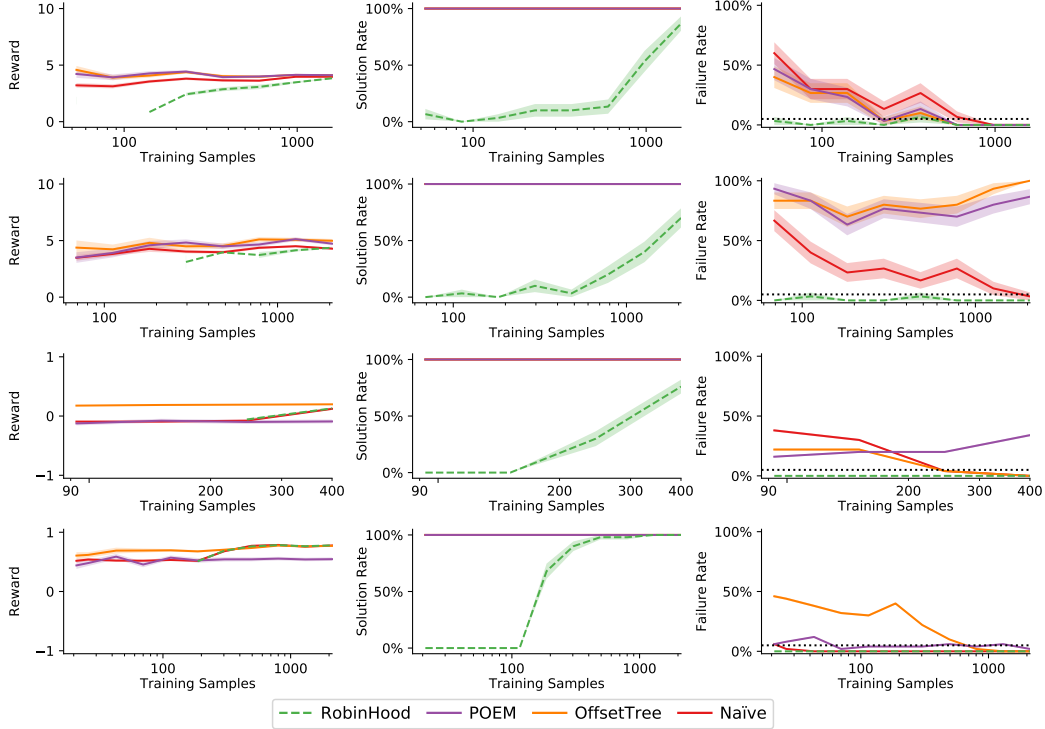

Figure 1: Each row presents results for different experiments generated over 30, 30, 50, and 50 trials respectively. Top row: tutoring sytsem, similar proportions with $\epsilon_m = 0.5$, $\epsilon_f = 0.0$. Second row: tutoring system, skewed proportions with $\epsilon_m = 0.5$, $\epsilon_f = 0.0$. Third row: enforcing disparate impact in the loan approval application with $\epsilon = -0.8$. Fourth row: enforcing statistical parity in the criminal recidivism application with $\epsilon = 0.1$. The dashed line denotes the maximum failure rate allowed by the behavioral constraints ($\delta = 5\%$ in our experiments).

computed, and the agent must trade-off between *exploiting*, i.e., taking actions that will maximize $\rho$, and *exploring*, i.e., taking actions it believes are suboptimal to build a better estimate of $\rho$ for taking that action. Most fairness-unaware algorithms eventually learn a policy that acceptably maximizes $\rho$, but there are no performance guarantees for policies between initial deployment and the acceptable policy, i.e., while the agent is exploring. These intermediate polices may choose suboptimal actions too often—this can be problematic in a real-world system, where choosing suboptimal actions could result in unintended inequity. In effect, fairness research for online methods has mostly focused on conservative exploration [24; 25; 35; 14; 27; 22]. The notion of action exploration does not apply in the offline setting because the agent does not interact iteratively with the environment. Instead, the agent has access to data collected using previous policies not chosen by the agent. Because of this, fairness definitions involving action exploration are not applicable to the offline setting.

Related work also exists in online metric-fairness learning [18; 41], multi-objective contextual bandits (MOCB) [45; 48], multi-objective reinforcement learning (MORL) [40; 49], and data-dependent constraint satisfaction [11; 47]. `RobinHood` can address metric-based definitions of fairness that can be quantitatively expressed using the set of operations defined in Section 4. MOCB and MORL largely focus on approximating the Pareto frontier to handle multiple and possibly conflicting objectives, though a recent batch MORL algorithm proposed by Le et al. [31] is an exception to this trend—this work focuses on problems of interest (with respect to fair policy learning) that can be framed as chance-constraints, and assumes the convexity of the feasible set $\Theta$. `RobinHood` represents a different approach to the batch MOCB setting with high probability constraint guarantees. The interface for specifying fairness definitions (presented in Section 4) makes `RobinHood` conceptually related to algorithms that satisfy data-dependent constraints [11]. In fact, `RobinHood` can more generally be thought of as an algorithm that ensures user-defined properties or behaviors with high probability, e.g., properties related to fairness. Finally, `RobinHood` belongs to a family of methods called *Seldonian* algorithms [47]. This class of methods satisfies user-defined safety constraints with high probability.

**Acknowledgments**

This work is supported by a gift from Adobe and by the National Science Foundation under grants no. CCF-1453474, IIS-1753968, CCF-1763423, and an NSF CAREER award to Brunskill. Research reported in this paper was sponsored in part by the Army Research Laboratory under Cooperative Agreement W911NF-17-2-0196. The views and conclusions contained in this document are those of the authors and should not be interpreted as representing the official policies, either expressed or implied, of the Army Research Laboratory or the U.S. Government.

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
