[Supplementary Material · fair_bandits_sup.pdf]

# A    Constructing Constraint Objectives for Common Fairness Definitions

This appendix provides examples of how to construct constraint objectives for several common fairness definitions. Our example domain will be the loan approval problem described in Section 2. In this problem, a bank is interested in maximizing the expected number of loan repayments it receives. The bank chooses to formulate this as an offline bandit problem such that, for each loan applicant, a single action, i.e., whether or not the applicant should be given a loan, is chosen.

We consider an action to belong to the *positive class* if the action corresponds to approving a loan, and we say it belongs to the *negative class* otherwise. We define an outcome to be whether or not an applicant repays (or would have repaid) a loan. We consider an outcome to belong to the positive class if the applicant repays the loan, and we say it belongs to the negative class otherwise. Many of the statistical measures of fairness we consider in this section rely on metrics we introduce below, framed in the context of the loan approval example.

- *True positive* (TP): the event in which the action chosen by the policy and the actual outcome both belong to the positive class. A true positive in the loan approval setting occurs when an applicant who would repay a loan is given a loan.

- *False positive* (FP): the event in which the action chosen by the algorithm is in the positive class when the actual outcome belongs to the negative class. A false positive in the loan approval setting is when an applicant who would not repay a loan is given a loan.

- *False negative* (FN): the event in which the action chosen by the policy is in the negative class but the actual outcome is in the positive class. A false negative in the loan approval setting is when an applicant who would have repaid a loan is denied a loan.

- *True negative* (TN): the event in which the action chosen by the policy and the actual outcome both belong to the negative class. A true negative in the loan approval setting is when an applicant who would not have repaid a loan is denied a loan.

Let TPR=TP/(TP+FN), FPR=FP/(TP+FN), FNR=FN/(TP+FN), and TNR=TN/(FP+TN) be the true positive, false positive, false negative, and true negative rates, respectively. Assume that the bank has unbiased estimators of these terms.

We now provide examples of how to construct objective constraints for some common definitions of fairness. In each definition, the bank is interested in guaranteeing fairness with respect to gender (the protected group), which we will assume to be binary in order to simplify notation.

**Predictive Equality [10].**    A policy exhibits predictive equality if FP rates are equal between groups. In our loan approval problem, this implies that the probability that an applicant who would not have repaid a loan be incorrectly approved for a loan should be the same for male- and female-identifying applicants. Predictive equality can be defined as $\mathbf{E}[\text{FPR}|\text{f}] = \mathbf{E}[\text{FPR}|\text{m}]$. To construct a constraint objective that satisfies $g(\theta) \leq 0$ if $\theta$ is fair, we can set $g = |\mathbf{E}[\text{FPR}|\text{f}] - \mathbf{E}[\text{FPR}|\text{m}]| - \epsilon$. This gives us a constraint objective for (approximate) predictive equality.

**Equal Opportunity [20].**    A policy exhibits equal opportunity if FN rates are equal between groups. In our loan approval problem, this implies that the probability that an applicant is denied a loan when they would have repaid it is equal between male- and female-identifying applicants. Equal opportunity can be defined as $\mathbf{E}[\text{FNR}|\text{f}] = \mathbf{E}[\text{FNR}|\text{m}]$. To construct a constraint objective that satisfies $g(\theta) \leq 0$ if $\theta$ is fair, we can set $g = |\mathbf{E}[\text{FNR}|\text{f}] - \mathbf{E}[\text{FNR}|\text{m}]| - \epsilon$. This gives us a constraint objective for (approximate) equal opportunity.

**Equalized Odds [20] (Conditional Procedure Accuracy Equality [5]).**    This definition combines predictive equality and equal opportunity: a policy exhibits equalized odds if FPR and TPR are equal between protected and unprotected groups. Assume the bank has unbiased estimators of FPR and TPR. Then equalized odds can be defined as $(\mathbf{E}[\text{FPR}|\text{m}] = \mathbf{E}[\text{FPR}|\text{f}]) \wedge (\mathbf{E}[\text{TPR}|\text{m}] = \mathbf{E}[\text{TPR}|\text{f}])$. To satisfy $g(\theta) \leq 0$ if $\theta$ is fair, we can set $g = |\mathbf{E}[\text{FPR}|\text{f}] - \mathbf{E}[\text{TPR}|\text{m}]| + |\mathbf{E}[\text{FPR}|\text{m}] - \mathbf{E}[\text{TPR}|\text{f}]| - \epsilon$.

**Treatment Equality [5].**    This definition focuses on the ratio of FN and FP errors for each group. A policy exhibits treatment equality if the ratio of FNs and FPs is equal for both female-

---
**Algorithm 3** ComputeUpperBounds($\theta, D, \Delta, \hat{Z}, \mathcal{E}, \texttt{inflateBounds}$)
---
1: $\texttt{out} = [\,]$
2: **for** $i = 1, ..., k$ **do**
3:     $\hat{Z}_i = \{\hat{z}_j^i\}_{j=1}^{d_i} \subseteq \hat{Z}$
4:     $L_i, U_i = \texttt{Bound}(E_i, \theta, D, \delta_i/d_i, \hat{Z}_i, \texttt{inflateBounds})$
5:     $\texttt{out.append}(U_i)$
6: **return** $\texttt{out}$
---

and male-identifying applicants. For the loan approval problem, this definition can be written as $(\mathbf{E}[\text{FN}|\text{f}]/\mathbf{E}[\text{FP}|\text{f}]) = (\mathbf{E}[\text{FN}|\text{m}]/\mathbf{E}[\text{FP}|\text{m}])$. For intuition, if the left-hand side of this expression were greater than the right-hand side, then either fewer female-identifying applicants were incorrectly assigned to the negative class than male-identifying applicants or more females-identifying applicants were incorrectly assigned to the positive class than male-identifying applicants. To construct a constraint objective that satisfies $g(\theta) \leq 0$ if $\theta$ is fair, we can set $g = |(\mathbf{E}[\text{FN}|\text{f}]/\mathbf{E}[\text{FP}|\text{f}]) - \mathbf{E}[\text{FN}|\text{m}]/\mathbf{E}[\text{FP}|\text{m}])| - \epsilon$.

# B   RobinHood Algorithm

This section elaborates on the helper methods (Algorithms 3 through 6) our approach uses. Recall that during candidate selection, $\hat{U}_i$ is calculated as the estimate of $U_i$, the upper bound of a behavioral constraint function $g_i$. This is done by substituting all quantities that depend on $D_s$ with approximations computed from $D_c$, and inflating the confidence intervals used to reflect this substitution, which is indicated by the Boolean variable $\texttt{inflateBounds}$. When set to $\texttt{True}$, the number of $g(\theta_c)$ estimates obtained using $D_s$ is approximated as $m = m|D_c|/|D_s|$ (Algorithm 5).

Algorithm 6 uses concentration inequalities [36] to produce high probability upper and lower bounds for a given behavioral constraint. It does this by recursively looking at sub-expressions of $E$ until a base variable is encountered, upon which, Algorithm 4 is called to calculate a real-valued high-confidence bound on the base variable's estimators. Correctness of Algorithm 6 is shown in Section C, and mostly follows from correct use of interval arithmetic and probability theory.

Different concentration inequalities can be substituted into Algorithm 5 to calculate upper and lower $(1-\delta_i)$-confidence bounds on each $g_i$. The pseudocode in Algorithm 5 presents both Hoeffding's inequality and Student's $t$-test as examples, where $\sigma$ is the sample standard deviation with Bessel's correction and $t_{1-\delta_i,m}$ is the $100(1-\delta_i)$ percentile of the Student's $t$ distribution with $m$ degrees of freedom. Note that Hoeffding's inequality often requires algorithms to be very conservative, i.e., high-probability guarantees may require an impractical amount of data. We could instead use Student's $t$-test, which is more data-efficient but assumes that the amount of many random variables is normally distributed. Due to the central limit theorem this is a reasonable assumption, but does not hold in general.

Finally, different policy evaluation methods, concentration inequalities, and optimization methods can be used for this algorithm. Here, we describe the specific modules we use in our experiments. Our experiments use *importance sampling* [39] to compute the expected return of a potential solution $\pi_c$. Next, we use several concentration inequalities to bound the base variables used in our algorithm. As mentioned above, Hoeffding's inequality or the Student's $t$-test were used for most experiments. In addition, we use confidence intervals based on Bootstrap methods [15] due to their desirable variance-reduction properties in the tutoring experiment. Lastly, $\texttt{RobinHood}$ uses a black box search method to solve the optimization problem on line 2 of Algorithm 1. In our experiments, we use the CMA-ES [19] implementation provided in the Python package cma (Python package).

Our algorithms were implemented in Python 3.6 (https://www.python.org/) using the numerical processing package, Numpy 1.15 (https://www.numpy.org/).

For the criminal recidivism and loan approval, a partition of $60\%$ of the total data was used for testing and $40\%$ for training. For the tutoring system experiments, we instead used $80\%$ of the data for training. In all experiments, when training RobinHood, $40\%$ of the training data was used for candidate selection and $60\%$ for the safety test.

**Algorithm 4** CIbound($\theta, D, \delta, \hat{Z}$, `inflateBounds`)

1: $z = \hat{z}_j(\theta, D)$
2: `margin` = CIFunction($z, D, \delta$, `inflateBounds`)
3: **return** $\Big(\text{mean}(z) - \texttt{margin}, \ \text{mean}(z) + \texttt{margin}\Big)$

---

**Algorithm 5** CIfunction($z, D, \delta$, `inflateBounds`)

1: `scale` $= 1$
2: $m = \text{length}(z)$
3: **if** `inflateBounds` **then**
4:     `scale` $= 2$
5:     $m = m\frac{|D_s|}{|D_c|}$
6: **case** `Hoeffding`: CI $= (b-a)\sqrt{\frac{\ln 1/\delta}{2m}}$     **case** `Student's` $t$: CI $= \frac{\sigma(z)}{\sqrt{m}}t_{1-\delta,m-1}$
7: **return** `scale` $\times$ CI

---

## C   Proof of Correctness for Recursive Bounds

In this section, we show that the upper bounds computed by the helper functions in Appendix B (Algorithms 3 through 6) satisfy $\Pr(g_i(\theta) > U_i) \leq \delta_i$ for all constraint objectives. This follows from the use of concentration inequalities and transformations to convert bounds on the base variables $z_j(\theta_c)$, into a high-confidence upper bound, $U_i$ on $g_i(\theta_c)$. More formally, we want to prove Lemma 1 below, whose assumptions are with respect to Hoeffding's inequality [21]. We show the proof for this lemma as well as a similar proof using Student's $t$-test [43] here.

**Lemma 1** (Recursive Bounds: Hoeffding's Inequality). *The upper bounds, $\{U_i\}_{i=1}^k$, returned by* `ComputeUpperBounds` *(Algorithm 3) satisfy the following inequality when* `inflateBounds=False` *and $\hat{z}_j(\theta_c, D)$ is bounded in some interval $[a, b]$ for $\theta_c$ and all $j \in \{1, ..., d\}$: $\forall i \in \{1, ..., k\} \Pr(g_i(\theta) > U_i) \leq \delta_i$.*

For Student's $t$-test, assume that $\{U_i\}_{i=1}^n$ satisfies Lemma 1 when `inflateBounds=False` and Average($\hat{z}_j(\theta_c, D)$) is normally distributed.

*Proof.* Assume $l_1$, $u_1$, and $l_2$, $u_2$ are $1-\delta/2$ confidence lower and upper bounds on base variables $z_1$ and $z_2$, respectively. Then $\Pr(z_1 \in [l_1, u_1]) \geq 1-\delta/2$ and $\Pr(z_2 \in [l_2, u_2]) \geq 1-\delta/2$. Below, we show that operations on base variables $z_1$ and $z_2$ result in high probability bounds. Note that throughout this proof, we make use of interval arithmetic and the following fact, which holds by the Union Bound:

$$\Pr(z_1 \in [l_1, u_1], z_2 \in [l_2, u_2]) \geq 1-\delta.$$

**Addition:** The event $(z_1 \in [l_1, u_1] \wedge z_2 \in [l_2, u_2])$ implies that $z_1 + z_2 \in [l_1 + l_2, u_1 + u_2]$. Then, $\Pr((z_1 + z_2) \in [l_1 + l_2, u_1 + u_2]) \geq \Pr(z_1 \in [l_1, u_1] \wedge z_2 \in [l_2, u_2])$, which is at least $1-\delta$. So, $\Pr((u_1 + u_2) \in [l_1 + l_2, u_1 + u_2]) \geq 1-\delta$.

**Maximum:** The event $(z_1 \in [l_1, u_1] \wedge z_2 \in [l_2, u_2])$ implies that $\max\{z_1, z_2\} \in [\max\{l_1, l_2\} \wedge \max\{u_1, u_2\}]$. Then, $\Pr(\max\{u_1, u_2\} \in [\max\{l_1, l_2\} \wedge \max\{z_1, z_2\}]) \geq \Pr(z_1 \in [l_1, u_1] \wedge z_2 \in [l_2, u_2])$, which is at least $1-\delta$. So, $\Pr(\max\{z_1, z_2\} \in [\max\{l_1, l_2\}, \max\{u_1, u_2\}]) \geq 1-\delta$.

**Product:** Let $A := \min\{l_1 l_2, l_1 u_2, l_2 u_1, l_2 u_2\}$ and $B := \max\{u_1 l_1, u_1 l_2, u_2 l_1, u_2 l_2\}$. The event $(z_1 \in [l_1, u_1] \wedge z_2 \in [l_2, u_2])$ implies that $(z_1 \times z_2) \in [A, B]$. Then, $\Pr((z_1 \times z_2) \in [A, B]) \geq \Pr(z_1 \in [l_1, u_1] \wedge z_2 \in [l_2, u_2])$ which is at least $1-\delta$. So, $\Pr((z_1 \times z_2) \in [A, B]) \geq 1-\delta$.

This can extend to $d$ base variables instead of two, in which case the operations on the probability inequalities described are less than or equal to $1-\delta/d$. Now, assume $l$ and $u$ are endpoints of an interval that satisfies $\Pr(z \in [l, u]) \geq 1-\delta$, for base variable $z$.

**Negation:** The event $z \in [l, u]$ implies that $-z \in [-u, -l]$. Then, $\Pr(-z \in [-u, -l]) \geq 1-\delta$.

**Algorithm 6** $\text{Bound}(E, \theta, D, \delta, \hat{Z}, \texttt{inflate})$

---

1: $X = \{\theta, D, \delta, \hat{Z}, \texttt{inflate}\}$
2: **switch** $E$ **do**
3:     **case** $u(\theta)$
4:         **return** $\texttt{CIBound}(X)$
5:     **case** $-E$
6:         **return** $-\texttt{Bound}(E, X)$
7:     **case** $E_l + E_r$
8:         $(L_l, U_l) = \texttt{Bound}(E_l, X)$
9:         $(L_r, U_r) = \texttt{Bound}(E_r, X)$
10:         **return** $(L_l + L_r, U_l + U_r)$
11:     **case** $E_l \times E_r$
12:         $(L_l, U_l) = \texttt{Bound}(E_l, X)$
13:         $(L_r, U_r) = \texttt{Bound}(E_r, X)$
14:         **return** $(\min\{L_l L_r, U_l L_r, L_l U_r, U_l U_r\}, \ \max\{L_l L_r, U_l L_r, L_l U_r, U_l U_r\})$
15:     **case** $E^{-1}$
16:         $(L, U) = \texttt{Bound}(E', X)$
17:         **if** $0 \in [l, u]$ **then return** $\texttt{NaN}$ **else return** $(1/U, 1/L)$
18:     **case** $|E|$
19:         $(L, U) = \texttt{Bound}(E, X)$
20:         **if** $0 \in [l, u]$ **then**
21:             **return** $(\min\{0, |L|, |U|\}, \max\{|L|, |U|\})$
22:         **else**
23:             **return** $(\min\{|L|, |U|\}, \max\{|L|, |U|\})$
24:     **case** $\max\{E_l, E_r\}$
25:         $(L_l, U_l) = \texttt{Bound}(E_l, X)$
26:         $(L_r, U_r) = \texttt{Bound}(E_r, X)$
27:         **return** $(\max\{L_l, L_r\}, \max\{U_l, U_r\})$

---

**Inverse** $(1/z)$**:**    Let $E$ be the event $z \in [l, u]$. If $l = 0$, then $E$ implies that $1/z \in [1/u, \infty]$, and if $u = 0$, then $E$ implies that $1/z \in [-\infty, 1/l]$. If $0 \in [l, u]$, then $E$ implies that $1/z \in ([-\infty, 1/l] \cup [1/u, \infty]) = [-\infty, \infty]$. In these cases, $\Pr(1/z \in [1/u, \infty]) \geq 1-\delta$ and $\Pr(1/z \in [-\infty, 1/l]) \geq 1-\delta$, respectively. If $0 \notin [l, u]$, $E$ implies that $1/z \in [1/u, 1/l]$, and $\Pr(1/z \in [1/u, 1/l]) \geq 1-\delta$.

**Absolute value:**    If $0 \in [l, u]$, then the event $z \in [l, u]$ implies $|z| \in [0, \max\{|l|, |u|\}]$. In this case, $\Pr(|z| \in [0, \max\{|L|, U\}]) \geq 1-\delta$. Otherwise, $z \in [l, u]$ implies $|z| \in [\min\{|l|, |u|\}, \max\{|l|, |u|\}]$, and $\Pr(|z| \in [\min\{|l|, |u|\}, \max\{|l|, |u|\}]) \geq 1-\delta$.      $\square$

While our set of operations does not include much more than simple arithmetic, interval methods can also be applied to functions with certain behaviors, e.g., functions with properties of monotonicity. This can be used to extend the set of operations allowed on base variables.

## D    Proof of Theorem 1: High Probability Fairness Guarantee

Consider the contrapositive formulation of behavioral constraint $i$, $\Pr(g_i(a(D)) > 0) \leq \delta_i$. With respect to this expression, $g_i(a(D)) > 0)$ implies that $a(D)$ is not NSF, which further implies that $U_i \leq 0$ for all $i$, and thus $\Pr(g_i(a(D)) > 0) = \Pr(g_i(a(D)) > 0, U_i \leq 0)$. Next, we use the fact that the joint event, $(g_i(a(D)) > 0, U_i \leq 0)$ implies the event, $(g_i(a(D)) > U_i)$:

$$\Pr(g_i(a(D)) > 0) \leq \Pr\left(g_i(a(D)) > U_i(\theta_c, D_s)\right).$$

Lastly, we note that $g(a(D)) > 0$ implies that a solution was returned—that is, $a(D) = \theta_c$:

$$\Pr(g_i(a(D)) > 0) \leq \Pr(g_i(\theta_c) > U_i).$$

Assumption 1 shows that for any fixed parameter vector, $\theta \in \Theta$, the upper bounds estimated by Algorithm 3 using $\texttt{inflateBounds=False}$ satisfy $\Pr(g_i(\theta) > U_i) \leq \delta_i$. Because $\theta_c \in \Theta$ for

any $D_c$, it follows from the substitution, $\theta = \theta_c$, that $\Pr(g_i(a(D)) > U_i) \le \delta_i$. This implies that $\Pr(g_i(a(D)) > 0) \le \delta_i)$, which further implies $\Pr(g_i(a(D)) \le 0) \ge 1 - \delta_i$, completing the proof.

## E   Proof of Theorem 2: Consistency Guarantee

Recall that the logged data, $D$, is a random variable. To further formalize this notion, let $(\Omega, \Sigma, p)$ be a probability space on which all relevant random variables are defined, and let $D_n : \Omega \to D$ be a random variable, where $D_n = D_c \cup D_s$. We will discuss convergence as $n \to \infty$. $D_n(\omega)$ is a particular sample of the entire set of logged data with $n$ trajectories, where $\omega \in \Omega$. Below, we present formal definitions and assumptions necessary for presenting our main result. To simplify notation, we assume that there exists only a single behavioral constraint and note that the extension of our proof to multiple behavioral constraints is straightforward.

**Definition 1.** *We say that a function $f : M \to \mathbb{R}$ on a metric space $(M, d)$ is piecewise Lipschitz continuous with Lipschitz constant $K$ and with respect to a countable partition, $\{M_1, M_2, ...\}$, of $M$ if $f$ is Lipschitz continuous with Lipschitz constant $K$ on all metric spaces in $\{(M_i, d)\}_{i=1}^{\infty}$.*

**Definition 2.** *If $(M, d)$ is a metric space, a set $X \subseteq M$ is a $\delta$-covering of $(M, d)$ if and only if $\max_{y \in M} \min_{x \in X} d(x, y) \le \delta$.*

Let $\hat{c}(\theta, D_c)$ denote the output of a call to `CandidateUtility`$(\Theta, D_c, \Delta, \hat{Z}, \mathcal{E})$ and let $c(\theta) :=$ $r_{\min} - g(\theta)$. Because we assume that there exists only a single behavioral constraint, the candidate utility function can be rewritten as `CandidateUtility`$(\Theta, D_c, \delta, \hat{Z}, E)$. That is, there is only a single threshold $\delta$ and a single analytic expression $E$. The next assumption ensures that $c$ and $\hat{c}$ are piecewise Lipschitz continuous. Notice that the $\delta$-covering requirement is straightforwardly satisfied if $\Theta$ is countable for $\Theta \subseteq \mathbb{R}^m$ for any positive natural number $m$.

**Assumption 1.** *The feasible set of policies, $\Theta$, is equipped with a metric, $d_\Theta$, such that for all $D_c(\omega)$ there exist countable partitions of $\Theta$, $\Theta^c = \{\Theta_1^c, \Theta_2^c, ...\}$ and $\Theta^{\hat{c}} = \{\Theta_1^{\hat{c}}, \Theta_2^{\hat{c}}, ...\}$, where $c(\cdot)$ and $\hat{c}(\cdot, D_c(\omega))$ are piecewise Lipschitz continuous with respect to $\Theta^c$ and $\Theta^{\hat{c}}$ respectively with Lipschitz constants $K$ and $\hat{K}$. Furthermore, for all $i \in \mathbb{N}_{>0}$ and all $\delta > 0$ there exist countable $\delta$-covers of $\Theta_i^c$ and $\Theta_i^{\hat{c}}$.*

Next we assume that for all $\theta \in \Theta$, the user-provided analytic expression $E$ is a continuous function of the base variables. With the exception of division, all operators discussed in Section 4 satisfy this assumption. In fact, this assumption is still satisfied for division when positive base variables are used in the denominator.

**Assumption 2.** *For all $\theta \in \Theta$, $g(\theta)$ is a continuous function of the base variables, $z_1(\theta), z_2(\theta), ..., z_d(\theta)$.*

Next we assume that a fair solution, $\theta^\star$, exists such that $g(\theta^\star)$ is not precisely on the boundary of fair and unfair. This can be satisfied by solutions that are arbitrarily close to the fair-unfair boundary.

**Assumption 3.** *There exists an $\epsilon > \xi$ and a $\theta^\star \in \Theta$ such that $g(\theta^\star) \le -\epsilon$.*

Next we assume that the sample off-policy estimator, $\hat{r}$, converges almost surely to $r$, the actual expected reward. This is satisfied by most off-policy estimators [46].

**Assumption 4.** $\forall \theta \in \Theta, \hat{r}(\theta, D_c) \xrightarrow{a.s.} r(\theta)$.

Lastly, we assume that the method used in `RobinHood` for constructing high-probability upper and lower bounds of a sample mean constructs confidence intervals that converge almost surely to the true mean, i.e., we assume that the confidence intervals on the base variables converge almost surely to the true base variable values for all solutions.. Hoeffding and Student's $t$-test are two example concentration inequalities that have this property: Hoeffding's inequality converges to the mean deterministically as $n \to \infty$, while the confidence interval used by Student's $t$-test converges almost surely to the mean assuming the standard deviation is finite (to see this, notice that the $t$ statistic is bounded and $1/\sqrt{n} \to 0$).

**Assumption 5.** *The confidence intervals on the base variables, $z_1(\theta), z_2(\theta), ..., z_d(\theta)$, converge almost surely to the true base variable values for all $\theta \in \Theta$.*

We prove Theorem 2 by building up properties that culminate with the desired result, starting with a variant of the strong law of large numbers:

**Property 1** (Khintchine Strong Law of Large Numbers). *Let $\{X_\iota\}_{i=1}^{\infty}$ be independent and identically distributed random variables. Then $(\frac{1}{n}\sum_{i=1}^{n}X_\iota)_{n=1}^{\infty}$ is a sequence of random variables that converges almost surely to $\mathbf{E}[X_1]$, if $\mathbf{E}[X_1]$ exists, i.e., $\frac{1}{n}\sum_{i=1}^{n}X_\iota \xrightarrow{a.s.} \mathbf{E}[X_1]$.*

*Proof.* See Theorem 2.3.13 of Sen and Singer [42]. □

In this proof, we consider the set $\bar{\Theta} \subseteq \Theta$, which contains all solutions that are not safe, and some that are safe but fall beneath a certain threshold: $\bar{\Theta} := \{\theta \in \Theta : g(\theta) > -\xi/2\}$. At a high level, we will show that the probability that the candidate solution, $\theta_c$, viewed as a random variable that depends on the candidate data set $D_c$, satisfies $\theta_c \notin \bar{\Theta}$ converges to one as $n \to \infty$, and then that the probability that $\theta_c$ is returned also converges to one as $n \to \infty$.

First, consider the confidence intervals produced for each base variable, $z_j$. Let $l_j(\theta, D_c)$ and $u_j(\theta, D_c)$ be the upper and lower confidence intervals on $z_j(\theta)$, respectively.

**Property 2.** *For all $\theta \in \Theta$, $l_j(\theta, D_c) \xrightarrow{a.s.} z_j(\theta)$ and $u_j(\theta, D_c) \xrightarrow{a.s.} z_j(\theta)$.*

*Proof.* Concentration inequalities construct confidence intervals around the mean by starting with the sample mean of the unbiased estimates (in our case, $\hat{z}(\theta, D_c)$) and then adding or subtracting a constant. While some concentration inequalities are naturally in this form, such as Hoeffding's inequality and Student's $t$-test, others can be restructured to produce this form. Thus, given Assumption 5, $l_j(\theta, D_c)$ and $u_j(\theta, D_c)$ can be be written as $\text{Average}(\hat{z}(\theta, D_c)) + Z_n$, where $Z_n$ is a sequence of random variables that converges (almost surely) to zero. Since $\text{Average}(\hat{z}(\theta, D_c)) \xrightarrow{a.s.} z_j(\theta)$ by Property 1, we therefore have that both $l_j(\theta, D_c) \xrightarrow{a.s.} z_j(\theta)$ and $u_j(\theta, D_c) \xrightarrow{a.s.} z_j(\theta)$. □

Next, we are interested in showing that the upper bound, $\hat{U}$, returned by Algorithm 3 converges to $g(\theta)$. To clarify notation, here we write $\hat{U}(\theta, D_c)$ to emphasize that $\hat{U}$ depends on the solution, $\theta$, and the data, $D_c$, passed to the off-policy estimator.

**Property 3.** *For all $\theta \in \Theta$, $\hat{U}(\theta, D_c) \xrightarrow{a.s.} g(\theta)$.*

*Proof.* We have from Property 2 that the confidence intervals on the base variables converge almost surely to $z_1(\theta), z_2(\theta), \ldots, z_d(\theta)$. Furthermore, by Assumption 2 we have that $g(\theta)$ is a continuous function of these base variables. Recall that $\hat{U}(\theta, D_c)$ is the upper bound produced by pushing the confidence intervals on the base variables though the analytic expression for $g(\theta)$. Since $g(\theta)$ is a continuous function of these base variables, $\hat{U}(\theta, D_c)$ is a continuous function of the confidence intervals on the base variables. So, by the continuous mapping theorem [2], $\hat{U}(\theta, D_c)$ converges almost surely to the value that it takes when applied to the converged values for the base variables, i.e., $g(\theta)$. □

Recall that we define $\hat{c}(\theta, D_c)$ as the output of Algorithm 2, i.e., $\texttt{CandidateUtility}(\Theta, D_c, \delta, \hat{Z}, E)$. Below, we show that given a fair solution $\theta^\star$ and data $D_c$, $\hat{c}(\theta^\star, D_c)$ converges almost surely to $r(\theta^\star)$, the expected reward of $\theta^\star$.

**Property 4.** $\hat{c}(\theta^\star, D_c) \xrightarrow{a.s.} r(\theta^\star)$.

*Proof.* By Property 3, we have that given Assumption 2, $\hat{U}(\theta^\star) \xrightarrow{a.s.} g(\theta^\star)$. By Assumption 3, we have that $g(\theta^\star) \le -\epsilon$. Now, let

$$A = \{\omega \in \Omega : \lim_{n \to \infty} U(\theta^\star, D_c(\omega)) = g(\theta^\star)\}. \tag{2}$$

Recall that $\hat{U}(\theta^\star, D_c) \xrightarrow{a.s.} g(\theta^\star)$ means that $\Pr(\lim_{n \to \infty} \hat{U}(\theta^\star, D_c) = g(\theta^\star)) = 1$. So $\omega$ is in $A$ almost surely, i.e., $\Pr(\omega \in A) = 1$. Consider any $\omega \in A$. From the definition of a limit and the previously established property that $g(\theta^\star) \le -\epsilon$, we have that there exists an $n_0$ such that for all $n \ge n_0$, the candidate utility function, Algorithm 2, will return $\hat{r}(\theta^\star, D_c)$ (this avoids the discontinuity of the $\texttt{if}$ statement in Algorithm 2 for values smaller than $n_0$).

Furthermore, we have from Assumption 4 that $\hat{r}(\theta^\star, D_c) \xrightarrow{\text{a.s.}} r(\theta^\star)$. Let

$$B = \{\omega \in \Omega : \lim_{n \to \infty} \hat{r}(\theta^\star, D_c(\omega)) = r(\theta^\star)\}. \tag{3}$$

From Assumption 4 we have that $\omega$ is in $B$ almost surely, i.e., $\Pr(\omega \in B) = 1$, and thus by the countable additivity of probability measures, $\Pr(\omega \in (A \cap B)) = 1$.

Consider now any $\omega \in (A \cap B)$. We have that for sufficiently large $n$, Algorithm 2 will return $\hat{r}(\theta^\star, D_c)$ (since $\omega \in A$), and further that $\hat{r}(\theta^\star, D_c) \to r(\theta^\star)$ (since $\omega \in B$). Thus, for all $\omega \in (A \cap B)$, the output of the candidate utility function converges to $r(\theta^\star)$, i.e., $\hat{c}(\theta^\star, D_c(\omega)) \to r(\theta^\star)$. Since $\Pr(\omega \in (A \cap B)) = 1$, we conclude that $\hat{c}(\theta^\star, D_c(\omega)) \xrightarrow{\text{a.s.}} r(\theta^\star)$. $\square$

We have now established that the candidate utility function converges almost surely to $r(\theta^\star)$ for the $\theta^\star$ assumed to exist in Assumption 3. We now establish a similar result for all $\theta \in \bar{\Theta}$—that the output of the candidate utility function converges almost surely to $c(\theta)$ (recall that $c(\theta)$ is defined as $r_{\min} - g(\theta)$).

**Property 5.** *For all $\theta \in \bar{\Theta}$, $\hat{c}(\theta, D_c) \xrightarrow{\text{a.s.}} c(\theta)$.*

*Proof.* By Property 3, we have that $\hat{U}(\theta, D_c) \xrightarrow{\text{a.s.}} g(\theta)$. If $\theta \in \bar{\Theta}$, then we have that $g(\theta) > -\xi/2$. We now change the definition of the set $A$ from its definition in the previous property to a similar definition suited to this property. That is, let:

$$A = \{\omega \in \Omega : \lim_{n \to \infty} U(\theta, D_c(\omega)) = g(\theta)\}. \tag{4}$$

Recall that $\hat{U}(\theta, D_c) \xrightarrow{\text{a.s.}} g(\theta)$ means that $\Pr(\lim_{n \to \infty} \hat{U}(\theta, D_c) = g(\theta)) = 1$. So, $\omega$ is in $A$ almost surely, i.e., $\Pr(\omega \in A) = 1$. Consider any $\omega \in A$. From the definition of a limit and the previously established property that $g(\theta) > -\xi/2$, we have that there exists an $n_0$ such that for all $n \geq n_0$ the candidate utility function will return $r_{\min} - \sum_{i=1}^{k} \max\{0, \hat{U}_i\}$. By the same argument, $\hat{U}(\theta, D_c(\omega)) \to g(\theta)$. So, for all $\omega \in A$, the output of the candidate utility function, $\hat{c}(\theta, D_c(\omega)) \to r_{\min} - g(\theta) - \xi$, and since $\Pr(\omega \in A) = 1$ we therefore conclude that $\hat{c}(\theta, D_c(\omega)) \xrightarrow{\text{a.s.}} c(\theta)$. $\square$

By Property 5 and one of the common definitions of almost sure convergence,

$$\forall \theta \in \bar{\Theta}, \forall \epsilon > 0, \Pr\left(\lim_{n \to \infty} \inf\{\omega \in \Omega : |\hat{c}(\theta, D_n(\omega)) - c(\theta)| < \epsilon\}\right) = 1.$$

Because $\Theta$ is not countable, it is not immediately clear that all $\theta \in \bar{\Theta}$ converge simultaneously to their respective $c(\theta)$. We show next that this is the case due to our smoothness assumptions.

**Property 6.** $\forall \epsilon' > 0, \Pr\left(\lim_{n \to \infty} \inf\{\omega \in \Omega : \forall \theta \in \bar{\Theta}, |\hat{c}(\theta, D_c(\omega)) - c(\theta)| < \epsilon'\}\right) = 1.$

*Proof.* Let $C(\delta)$ denote the union of all the points in the $\delta$-covers of the countable partitions of $\Theta$ assumed to exist by Assumption 1. Since the partitions are countable and the $\delta$-covers for each region are assumed to be countable, we have that $C(\delta)$ is countable for all $\delta$. Then for all $\delta$, we have convergence for all $\theta \in C(\delta)$ simultaneously:

$$\forall \delta > 0, \forall \epsilon > 0, \Pr\left(\lim_{n \to \infty} \inf\{\omega \in \Omega : \forall \theta \in C(\delta), |\hat{c}(\theta, D_c(\omega)) - c(\theta)| < \epsilon\}\right) = 1. \tag{5}$$

Now, consider a $\theta \notin C(\delta)$. By Definition 2 and Assumption 1, $\exists\, \theta' \in \bar{\Theta}_i^c, d(\theta, \theta') \leq \delta$. Moreover, because $c$ and $\hat{c}$ are Lipschitz continuous on $\bar{\Theta}_i^c$ and $\bar{\Theta}_i^{\hat{c}}$ (by Assumption 1) respectively, we have that $|c(\theta) - c(\theta')| \leq K\delta$ and $|\hat{c}(\theta, D_c(\omega)) - \hat{c}(\theta', D_c(\omega))| \leq \hat{K}\delta$. So, $|\hat{c}(\theta, D_c(\omega)) - c(\theta)| \leq |\hat{c}(\theta, D_c(\omega)) - c(\theta')| + K\delta \leq |\hat{c}(\theta', D_c(\omega)) - c(\theta')| + \delta(K + \hat{K})$. This means that for all $\delta > 0$:

$$\left(\forall \theta \in C(\delta), |\hat{c}(\theta, D_c(\omega)) - c(\theta)| < \epsilon\right) \implies \left(\forall \theta \in \bar{\Theta}, |\hat{c}(\theta, D_c(\omega)) - c(\theta)| < \epsilon + \delta(K + \hat{K})\right).$$

Substituting this into (5), we get:

$$\forall \delta > 0, \forall \epsilon > 0, \Pr\left(\lim_{n \to \infty} \inf\{\omega \in \Omega : \forall \theta \in \bar{\Theta}, |\hat{c}(\theta, D_c(\omega)) - c(\theta)| < \epsilon + \delta(K + \hat{K})\}\right) = 1.$$

Now, let $\delta := \epsilon/(K + \hat{K})$ and $\epsilon' = 2\epsilon$. Thus, we have the following:

$$\forall \epsilon' > 0, \Pr\left(\lim_{n\to\infty} \inf\{\omega \in \Omega : \forall\theta \in \bar{\Theta}, |\hat{c}(\theta, D_c(\omega)) - c(\theta)| < \epsilon'\}\right) = 1.$$

$\square$

So, given the appropriate assumptions, for all $\theta \in \bar{\Theta}$, we have that $\hat{c}(\theta, D_c(\omega)) \xrightarrow{\text{a.s.}} c(\theta)$ and that $\hat{c}(\theta^\star, D_c(\omega)) \xrightarrow{\text{a.s.}} r(\theta^\star)$. Due to the countable additivity property of probability measures and Property 6, we have the following:

$$\Pr\left(\left[\forall\theta \in \bar{\Theta}, \lim_{n\to\infty} \hat{c}(\theta, D_c(\omega)) = c(\theta)\right], \left[\lim_{n\to\infty} \hat{c}(\theta^\star, D_c(\omega)) = r(\theta^\star)\right]\right) = 1, \qquad (6)$$

where $\Pr(A, B)$ denotes the joint probability of $A$ and $B$.

Let $H$ denote the set of $\omega \in \Omega$ such that (6) is satisfied. Note that $r_{\min}$ is defined as the value always less than $r(\theta)$ for all $\theta \in \Theta$, and $-g(\theta) \leq 0$ for all $\theta \in \bar{\Theta}$. So, for all $\omega \in H$, for sufficiently large $n$, the candidate utility function will not define $\theta_c$ to be in $\bar{\Theta}$. Since $\omega$ is in $H$ almost surely ($\Pr(\omega \in H) = 1$), we therefore have that $\lim_{n\to\infty} \Pr(\theta_c \notin \bar{\Theta}) = 1$.

The remaining challenge is to establish that, given $\theta_c \notin \bar{\Theta}$, the probability that the safety test returns NSF converges to zero as $n \to \infty$. By Property 3 (but using $D_s$ in place of $D_c$), we have that $\hat{U}(\theta_c, D_s) \xrightarrow{\text{a.s.}} g(\theta_c)$. Furthermore, we have that for all $\omega \in H$, there exists an $n_0$ such that for all $n \geq n_0$, $\theta_c \notin \bar{\Theta}$. From the definition of $\Theta$, this means that there exists an $n$ such that for all $n \geq n_0$, $g(\theta_c) < -\xi/2$, at which point the safety test would return $\theta_c$. So, we have our desired result—the limit as $n \to \infty$ of the probability that RobinHood returns a solution, $\theta_c$, is one, meaning the limit as $n \to \infty$ of the probability that RobinHood returns NSF is zero.

# F   Additional Experimental Details

This section provides pseudocode for the helper methods in RobinHood, details for the baselines, more details on our application domains, and a discussion of the results in our loan approval experiments.

## F.1   Baseline Methods

Except for NaïveFairBandit, the baselines we compare to exist as repositories online and are linked below.

- Offset Tree [6]: https://github.com/david-cortes/contextualbandits
- POEM [44]: http://www.cs.cornell.edu/ adith/POEM/
- Rawlsian Fair Machine Learning for Contextual Bandits [23]: An online contextual bandit algorithm which enforces *weakly meritocratic fairness* at every step of the learning process. We used the repository located at https://github.com/jtcho/FairMachineLearning for this experiment, but made changes to the original code. This was done to accurately reflect the original work in Joseph et al. [23]. Specifically, the repository code showed the bandit the reward for each action at every round, and this was instead changed to returning the reward for only the action chosen by the algorithm.
- NaïveFairBandit does not employ a safety test, and therefore does not search for candidate solutions with inflated confidence intervals or return NSF. Pseudocode for NaïveFairBandit is located in Algorithm 7.

## F.2   Tutoring Systems

**User Study.**   We define a new mathematical operator called the \$ operator such that $A\$B = B \times \lceil A/10 \rceil$. In total, three tutorial versions are used in our experiments. The \$ operator is described in different ways in each of the three tutorials, as depicted in Figures F.1 through F.3. Tutorial 1 (Figure F.1) describes the \$ operator using code, and includes example problems. Tutorial 2 (Figure F.2 describes the \$ operator in a non-intuitive way, and uses fewer example problems.

**Algorithm 7** `NaïveFairBandit`

1: **function** NAIVEFAIRBANDIT($D, \Delta, \hat{Z}, \mathcal{E}$)
2:     $\theta_c = \underset{\theta \in \Theta}{\arg\max} \texttt{ NaiveCandidateValue}(\theta, D, \Delta, \hat{Z}, \mathcal{E})$

3:
4: **function** NAIVECANDIDATEVALUE($\theta, D_c, \Delta, \hat{Z}, \mathcal{E}$)
5:     $[\hat{U}_1, ..., \hat{U}_k] = \texttt{ComputeUpperBounds}(\theta, D_c, \Delta, \hat{Z}, \mathcal{E}, \texttt{False})$
6:     $r_{\min} = \underset{\theta' \in \Theta}{\min} \hat{r}(\theta', D_c)$

7:     **if** $\hat{U}_i \leq 0$ for all $i \in \{1, ..., k\}$ **then return** $\hat{r}(\theta, D_c)$ **else return** $r_{\min} - \sum_{i=1}^{k} \max\{0, \hat{U}_i\}$

Tutorial 1

**What is the \$ Operator?**

Similar to addition and subtraction, the \$ operator requires two numbers as input. It is denoted by `dollar(A,B)` or A \$ B, for integers A and B.

The equation for \$ is below:

```
int dollar(int A, int B){
    float Y = 10;
    int C = ceil(A/Y);
    return B*C;
}
```

Figure F.1: Tutorial that defines the \$ operator using code.

Tutorial 3 (Figure F.3) responds differently to users based on their gender identification. Male-identifying users were given the straightforward definition of the \$ operator, while female-identifying users were given an incorrect definition. The number of female-identified users in this data set is 1,403 and the number of male-identified users is 1,178. In the equal proportions experiment, two tutorials were used for a total of 1988 samples. In the skewed proportions experiment, three tutorials were used for a total of 2581 samples were used. The assessment shown to each user once they completed the provided tutorial consists of ten questions using the \$ operator, e.g., 16\$4. When computing importance weights, we re-scale female and male samples to simulate a similar data set containing 80% males and 20% females. Response format was fill-in-the-blank.

### F.3   Loan Approval

**Statistical Parity.** In this experiment, we enforce statistical parity. A policy is considered fair in this case if its probability of approving male applicants is equal to its probability of approving female applicants. We use the `Personal Status and Sex` feature of each applicant to determine sex. Statistical parity can be encoded as the following constraint objective:

$$g(\theta) := \left| \mathbf{E}[A|\mathtt{m}] - \mathbf{E}[A|\mathtt{f}] \right| - \epsilon, \tag{7}$$

where $A = 1$ if the corresponding applicant was granted a loan and $A = 0$ otherwise. To satisfy this constraint objective, the absolute difference between the conditional expectations must be less than $\epsilon$ with high probability.

**Statistical Parity Experimental Results.** We ran 50 trials for this experiment. We find that for larger training set sizes the estimated expected reward of solutions returned by `RobinHood` grows steadily more comparable to those returned by Offset Tree and `NaïveFairBandit`. Very quickly, we see that baselines begin to return solutions that are fair with respect to each constraint objective, indicating that solutions that optimize performance also are in line with the behavioral constraints. As can be seen in Figure F.4, `RobinHood` is able to find and return these solutions as well.

Tutorial 2

### What is the $ Operator?

For this tutorial, we have invented the $ operator. Similar to addition and subtraction, the $ operator requires two numbers as input. It is denoted by A $ B, for integers A and B.

A $ B is equivalent to the expression:

$$B \times (Y + X)$$

where Y is equivalent to $\left\lceil \frac{A}{10} \right\rceil - 1$ if A is not divisible by 10 and Y is equivalent to $\left\lceil \frac{A}{10} \right\rceil$ otherwise; $\lceil \cdot \rceil$ represents the ceiling operator; and X is equivalent to 0 if A is divisible by 10 and X is equivalent to 1 otherwise.

Figure F.2: Tutorial that explains the $ operator in a convoluted manner.

Tutorial 3

### What is the $ Operator?

The equation for A $ B is below. You may want to write this down.

$$A\$B = B \times \left\lceil \frac{A}{10} \right\rceil$$

### What is the $ Operator?

Similar to addition and subtraction, the $ operator requires two numbers as input. It is denoted by A $ B, for integers A and B.

The equation for A $ B is as follows:

$$A\$B = B - \left\lceil \frac{A}{10} \right\rceil$$

Figure F.3: Biased tutorial that provides incorrect information to female-identifying users.

**Comparison to Online Fairness Methods.** Because most existing work on fairness for bandits considers the online setting, we performed an additional experiment to evaluate how RobinHood performs in comparison to these baselines. In particular, we compare RobinHood to the framework discussed by Joseph et al. [23], called Rawlsian Fair Machine Learning. This framework considers the contextual bandit setting in a slightly different way, where contexts are actions from which to choose. Importantly, we note that this experiment is not entirely fair, as RobinHood is formulated for the offline bandit setting. Nonetheless, the results of this experiment provide a rudimentary analysis of how our method might perform if used in this setting.

To compare our method, which learns using a batch of data, to baselines that make decisions online, we use the following procedure. First, we assume that the problem definition provides a reward function, $r : (\mathcal{X} \times \mathcal{A}) \to \mathbb{R}$, which determines the value of taking each action given a particular context. In our experiment, we obtain $r$ by training a Gaussian process regression model to predict $R$ given $(X, A)$, using the data from the loan approval experiment. Importantly, because this experiment uses a simulator, which only roughly approximates the processes that generated the loan approval dataset, we expect the results to differ from those that were obtained in the offline loan approval experiments. Given $r$, we train RobinHood as in offline experiments, with the exception that reward is computed dynamically using $r$ instead of being estimated using importance sampling. To train the online baseline algorithms, we allow them to iteratively learn over each context in the training data set in random order. As a result, despite the fact that RobinHood learns in batch while the baselines learn iteratively, our training procedure ensures that all models are trained on the same amount of data, and using the same set of contexts and reward function, $r$. Once the algorithms are trained, their parameters are fixed and they are evaluated on the remaining testing data. As in the offline

Figure F.4: Enforcing statistical parity in the Loan Approval domain ($\epsilon = 0.23$).

Figure F.5: Comparison to online methods for loan approval using disparate impact ($\epsilon = -0.8$).

experiments, we report evaluation statistics averaged over several randomized trials, where each trial uses 40% of the data for training and 60% for testing.

Figure F.5 shows the results of this experiment, which we averaged over 50 trials. Similar to our other results, `RobinHood` maintains an acceptable failure rate ($5\%$ in this experiment), and returns solutions other than `NSF` given a reasonable amount of data.

# G  Example Base Variable Without Unbiased Estimator

In Section 4 we mentioned that there may be some base variables that the user wants to use when defining fairness that do not have unbiased estimators (e.g., standard deviation). For example, say the user wanted to define a base variable to be the largest possible expected reward for any policy with parameters $\theta$ in some set $\Theta$, i.e., $\max_{\theta \in \Theta} r(\theta)$. That is, this base variable is: $z(\theta) \coloneqq \max_{\theta' \in \Theta} r(\theta')$. Note that because this quantity does not depend on the solution being evaluated, $\theta$, we denote it as $z$ instead of $z(\theta)$. This base variable is important in the context of this paper because it can be a useful component in definitions of fairness.

Consider an example application where each context corresponds to a person, actions correspond to deciding which tutorial on consumer economics to give to a person, and the reward is the person's fiscal savings during the following year. In this case, we might desire a system that is fair with respect to people from different locations, say Maryland and Mississippi. Since the mean income in Maryland in 2015 was $\$75{,}847$, while the mean income in Mississippi in 2015 was only $\$40{,}593$, it would not be reasonable to require the bandit algorithm to ensure that it selects actions in a way that ensures the expected return for people in both states is similar (the expected yearly income). That is, a tutoring system on resource economics could not be expected to remedy the income disparity between these two states.

Rather than compare the expected rewards for people in each state, it would be more reasonable to compare how far the expected reward is from the best possible expected reward for people in each state. This allows for behavioral constraints that require the expected yearly income (expected return) for people in each state to be within $\$500$ of the maximum possible expected income when considering the impact of different policies (tutoring systems). Alternatively, one might require the expected yearly income to be within $10\%$ of the best possible (our of all of the tutoring systems considered) for people in each state. To use these definitions of fairness, we might desire a base variable that is equal to the maximum possible expected reward for people of a particular type. The base variable that we present here shows how this can be achieved (although we present the variable without conditioning on a person's type, this extension is straightforward).

The challenge when upper-bounding $z$ is that we do not know which policy is optimal. It is straightforward to construct high-confidence upper bounds on the performance of a particular policy using importance sampling and a concentration inequality like Hoeffding's inequality. In this section only, let $U(\theta, D)$ denote a $(1-\delta)$-confidence upper bound on $z$, constructed from data, $D$. One approach to upper bound $z$ would be to first produce a high-confidence upper bound, $U(\theta', D)$, on the performance of each solution $\theta' \in \Theta$, and then compute the bound on $z$ using the set of upper bounds, $\{U(\theta', D)\}_{\theta' \in \Theta}$. Because $\theta^\star$ is the optimal policy with respect to $r$, i.e.,

$$\theta^\star \in \arg\max_{\theta' \in \Theta} r(\theta'), \tag{8}$$

one candidate for a $(1-\delta)$-confidence upper bound on $z$ is the supremum of the upper bounds computed for each $\theta'$: $\sup_{\theta' \in \Theta} U(\theta', D)$.

However, it is not clear that this is a valid upper bound on $z$. Intuitively, if the upper bound computed for *any* of the $\theta' \in \Theta$ is too small, then this proposed upper bound on $r(\theta^\star)$ may be too small. So, it may seem that the probability that this upper bound fails is the probability that there is one or more $\theta' \in \Theta$ for which $U(\theta', D)$ does not upper bound $z$. If $\Theta$ has cardinality $n$, then this probability can be as large as $n\delta$. Here we show that this reasoning is incorrect: for this particular base variable, the probability of failure remains $\delta$, not $n\delta$. That is:

**Property 7.** *Let $\Theta$ be a (possibly uncountable) set of policy parameters (policies). If, there exists at least one $\theta^\star$ satisfying* (8) *and for all $\theta' \in \Theta$, $U(\theta', D)$ is a $(1-\delta)$-confidence upper bound on $r(\theta')$, constructed from data $D$, then*

$$\Pr\left(\sup_{\theta' \in \Theta} U(\theta', D) \geq \max_{\theta' \in \Theta} r(\theta')\right) \geq 1-\delta. \tag{9}$$

*Proof.* Note that in (9), $U(\theta', D)$ is the only random variable—it is the source of all randomness. Furthermore, by the assumption that there exists at least one $\theta^\star$ satisfying (8), $\max_{\theta' \in \Theta} r(\theta')$ exists. However, $\max_{\theta' \in \Theta} U(\theta', D)$ may not exist, and so we use $\sup_{\theta' \in \Theta} U(\theta', D)$.

Let $\theta^\star$ be any policy satisfying (8). By assumption, $U(\theta^\star)$ is a $(1-\delta)$-confidence upper bound on $r(\theta^\star)$. That is,

$$\Pr\left(U(\theta^\star, D) \geq r(\theta^\star)\right) \geq 1-\delta. \tag{10}$$

Because $\theta^\star$ satisfies (8), we have that $r(\theta^\star) = \max_{\theta' \in \Theta} r(\theta')$, and so

$$\Pr\left(U(\theta^\star, D) \geq \max_{\theta' \in \Theta} r(\theta')\right) \geq 1-\delta. \tag{11}$$

Since $\theta^\star \in \Theta$, we have that $U(\theta^\star, D) \leq \sup_{\theta' \in \Theta} U(\theta', D)$, and so

$$\Pr\left(\sup_{\theta' \in \Theta} U(\theta', D) \geq \max_{\theta' \in \Theta} r(\theta')\right) \geq 1-\delta. \tag{12}$$

$\square$