[Reviews · NeurIPS 2019]

Reviewer 1



- An important meta-observation is that the authors focus their work to be contributions to “fairness”. The reviewer believes that the paper is indeed motivated by discussions of fairness, however the paper leans too much on fairness wording, while the actual contributions are about making sure a model adheres to statistical properties and providing tools for developers to check for these. Specific fairness metrics are then examples where this approach can be used. To be more specific, the word “fair” is problematic in the conclusion that “RobinHood is able to return fair solutions with high probability given a reasonable amount of data”. Whether a solution is fair depends on many more aspects than just the statistics of the model, including the context in which a model is used and the ways in which the model is built. That said, there is great value in making sure models are statistically sound to begin with. And it is this basic but crucial challenge that your paper is taking up. - It is noted that the authors require 11 pages of supplemental material. As a result, both the algorithmic section as well as the theoretical section in the main paper are made rather brief and impossible to understand/reproduce without the appendices. The reviewer does not know what the Neurips standards are exactly, but this feel to be a bit over the manageable amount of supplemental material for rigorous review and reproducibility purposes. If the authors have ways to reduce this material and improve the flow of the paper (algorithmic and theoretical sections), that would be appreciated. Concretely, the reviewer believes it’s better to introduce all concepts that are mentioned in the paper in the main paper and not in the appendices (examples: inflateBounds, Assumptions of Theorem 2). - How is the candidate selection done? If the candidate set does not work, how do you resample in a way that increases the likelihood of success? In other words, what is the algorithm to optimize over different candidate sets? This should be addressed in the text. - In Algorithm 1, how do you update the iteration, or how do you construct the set \Theta? Is this agnostic to the ML model class/algorithm used? - The following sentence lacks clarity: “At a high level, in these helper functions, bounds on the base variables are found by recursively looking at sub-expressions of $E_i$ until a base variable is encountered, upon which, real valued upper and lower $(1 - \delta_i)$-confidence bounds on the base variable’s estimators are computed and subsequently propagated through $E_i$”. Please explain what is meant by “sub-expressions of $E_i$” and “propagated through $E_i$”. - Lemma 1 requires on details of algorithms specified in Appendix (inflateBounds) to set up the lemma. The reviewer is not fully familiar with the standards for Neurips papers, but it is advised to write the Lemma in plain English and only use quantities that are introduced in the main text. - Line 206: The term Seldonian does not seem to be very common. Please explain what it entails. - Theorem 2: the Assumptions are all written in the appendix. If the theorem is important, then understanding under what assumptions it holds, and what these assumptions mean in a practical context is just as important. Therefore the authors are advised to render visible in the main paper the assumptions (and thereby the affordances and restrictions) of their approach. - Line 250: in claiming the ability to “satisfy multiple and novel fairness criteria”, please provide nuances about the limitations of your approach. There are inherent impossibilities in trying to satisfy multiple criteria, see [1, Section “Formal non-discrimination criteria”] and [2]. The paper’s algorithmic advance does not undo these formal restrictions, and the authors should there be careful about their claims, doing justice to the inherent limitations of fairness work, also acknowledging the . [1] S. Barocas, M. Hardt, and A. Narayanan, “Fairness in machine learning,” NIPS Tutorial, 2017. [2] J. Kleinberg, S. Mullainathan, and M. Raghavan, “Inherent Trade-Offs in the Fair Determination of Risk Scores,” in 8th Innovations in Theoretical Computer Science Conference (ITCS 2017), Dagstuhl, Germany, 2017, vol. 67, pp. 43:1--43:23. - Line 254: It is not clear what the role of expected rewards for female and male are in determining whether a model is fair. In line 261, fairness is mentioned without a proper explanation of why that is fair. It seems to the reviewer that there may be structural issues in the expected rewards, and therefore it would be needed to be more clear and explicit about why these constraint objectives are chosen. If it is just about making sure the empirical mean and expected value are in line for each group, that is merely making sure that each group is well trained, and the language of “fairness” should be toned down. - Equation below line 296 seems to have a typo P(A|C) = 1, should probably be P(A=1|C). Same for the other probability. - The related work session comes at the end. The authors are advised to move it to the beginning of the paper. It is important for motivating the work in this paper. For instance, in lines 330-334: The authors state a high-level reason why “the prior classification-centered approaches cannot directly apply due to the different type of environmental feedback in these settings”. This is an important argument, which should be part of the introduction. In addition, the authors should explain which fairness metrics can and cannot be applied and make clearer why (the feedback argument is not clear enough).

Reviewer 2



Overall Comments This paper tackles an ambitious question of providing an algorithm in the offline contextual bandit setting that satisfies fairness guarantees that are specified as constraints. The paper provides a high probability guarantee for this algorithm and also empirically tests the algorithm on a variety of different data sets. The paper seeks an algorithm that satisfies a few conditions: 1) that the fairness constraints be user specified, 2) that a solution is returned if one exists 3) and that the 'sample complexity' of this algorithm is 'reasonable'. The combination of these contributions make this a high quality submission in my opinion. Areas for improvement are noted below. Originality As far as I can tell, this is the first offline contextual bandit algorithm that satisfies a user defined fairness constraints. In general, the goal of developing algorithms that satisfy user defined fairness definitions is not new, however, the formulation in this paper is new, challenging, and interesting in the offline batch setting. Previous work, see Cotter et. al. 2018ab, noted below has focused on developing algorithms that include rate constraints as part of the training process. In principle, the fairness constraints presented here are conceptually in this family. Quality Overall, this is a high quality paper that tries to solve an important problem in a comprehensive manner. As with any paper, it does not solve all of the problems and leaves open the question about the exact sample complexity of the proposed algorithm; however, the paper demonstrates empirically that it is likely similar to other proposed methods. Clarity Overall this paper is clear and relatively easy to follow. My biggest suggestion for improvement is in section 5 (theoretical analyses section). In section 5, it would have helped substantially to provide a proof sketch or general overview of how and why assumptions 1-5 are necessary for theorem 2. Most of this is left to the appendix, which is fine, but since these proofs are a key contribution of this work, doing more to set them up would have been great. Significance This work is significant since it moves the area towards providing algorithms that help satisfy fairness constraints while also providing the option for providing answers like 'I don't know' or 'No solution found'. The paper also demonstrates the utility of the proposed Robinhood algorithm in a 3 settings that are significant. Minor Fixes Page 8, figure 1: Include a caption and figure label. In addition, can you improve on the color selection for some of the algorithms ? It is currently difficult to tell apart POEM vs OffsetTree since the lines are both red. l296: Include brackets in this equation, right now it is somewhat difficult to read. l422: Citation 29 is empty. Areas for Possible Improvement/Questions Here are some questions that the authors could possibly clarify about this paper. 1) The guarantee that is given in the high probability proofs are mostly because of the iid assumptions that the authors make. One of the exciting things about the contextual bandit framework/bandits in general is that one can do away with this assumption. What are the potential avenues one has for giving this kind guarantee for a non-iid setting? Perhaps it makes sense to require this because the authors are operating in the batch setting here. 2) Presentation of theorem 2 in the appendix. I spent quite a bit of time with this theorem, however, it is still difficult to understand the importance of the different pieces that the paper uses to make this claim. Ultimately, I couldn't spot an error, but I would've liked the authors to clarify why each property/assumption is needed for their claim. Some are obvious like the use of the hoeffding's inequality. Assumption 3 for example, indicates that a fair solution, for a constraint, $g(\theta^\ast)$ exists. This seems to me like a very strong assumption that should be relaxed somewhat. Ultimately, more clarification in these portion would help the reader. 3) Figure 1 does not include the loan approval figures. 4) Column 3 in the figures is interesting; it suggests that as you train the baseline algorithms that the paper compares against with more samples, these algorithms start to approach failure that is close to the proposed robinhood approach. Can the authors help provide insight to these finding? 5) Demographic parity/Statistical Parity. As I am sure the authors are aware, there are other statistical definitions like equalized odds and other variants. Is it straightforward to extend this to those definitions provided an appropriate $g$ is given? Update I have read the author rebuttal, and will maintain my rating. The plan to provide clarification for section 5 in the final version. I consider to work to be sound and addressing a very challenging question.

Reviewer 3



The paper proposes a novel problem in the area of counterfactual learning. None of the previous work has explored this. The paper would be a good fit for the conference and a good contribution to the area of machine learning given the authors can fix a few minor issues with the explanations and text. The paper is well written with only a few typos. Most of the results and experiment setups are clear. Proofs are present in the appendix but the major content is still included in the main text. 1) I really like the idea of certifying fairness through a held out safety dataset and suggesting that no solution exists and more data needs to be collected to find a feasible fair solution. 2) I am still not sure about the name RobinHood maybe this needs some explanation? 3) There is no mention of the support assumption that offline CB works usually make i.e. whether we can even evaluate a policy offline with the given data under the logging policy h_\theta. I am assuming it will be a part of the CandidateUtility function (or the argmax thereafter) but should it make a difference? if yes, in what way? 4) There are simpler baselines that the paper could have compared to. For example, one could use a Lagrangian multiplier to impose the same fairness constraint g_theta while learning using gradient descent on an IPS/SN-IPS estimator (Swaminathan&Joachims). This would already do pretty well in my opinion. (This could be Naive FairBandit but algo 7 seems missing from the appendix). 5) I have not carefully gone through the proofs. But I would suggest that the paper contains a description of the inflateBounds argument in the ComputeUBs method. Why is it needed to compute the candidate utility but not when certifying fairness? 6) A uniform random policy would be a fair one (for example in case of the Tutoring system example Section 6.1). Wouldn't it be ideal to return that policy as the solution rather than an NSF? Or even better, if we can find a suboptimal utility policy that satisfies the fairness criterion, shouldn't it be the idea to return that. Also, is there a way to say if given an estimator for g_theta, there might not exist such a policy? For an equal opportunity kind of constraint, my opinion is that the uniform random policy always satisfies the fairness constraint. 7) Some minor clarifications and typos: - line 252: Is the expectation E on the policy \pi_theta - eqn in line 296: it should be P(A=1|caucasian)... rather than P(A|Caucasian)=1

[Author Response · NeurIPS 2019]

We thank the reviewers for their insightful responses. Due to space limitations, we were unable to respond to all of the
comments we found valuable, e.g., properly defining the term 'Seldonian', strengthening the introduction with material
from the related work, properly describing the recursive process of computing bounds on terms in the expression $E$,
potential avenues for giving high probability guarantees for a non-iid setting, etc. We will incorporate this feedback.

Reviewers suggested moving material from the supplemental section into the main body, such as an explanation of
theoretical assumptions, more detailed algorithm descriptions, introduction of quantities used in lemmas and theorems,
and additional experimental figures. We will do our best to incorporate this material in the main body. If accepted,
NeurIPS allows an additional page in the main body, which will help us to do this, along with massaging existing text.

**R1: Extending to other statistical definitions, like equalized odds and other variants.** RobinHood applies to all
definitions that can be represented as certain operations (listed in Section 4) on variables for which high-confidence
upper and lower bounds can be computed. This includes variables with unbiased estimators, e.g., false positive rates
(FPR) and true positive rates (TPR), and variables without unbiased estimators, e.g., standard dev. We will make a point
to elaborate on how this can be extended to other statistical definitions in the main text. As an example of how to create a
behavioral constraint that enforces (approximate) equalized odds in the loan approval problem, we assume that the user
has unbiased estimators of TPR and FPR. Equalized odds requires that FPR and TPR are equal between protected and
unprotected groups. To satisfy $g(\theta) \leq 0$ if $\theta$ is fair, we can set $g = |\mathbf{E}[\text{FPR}|\mathtt{f}] - \mathbf{E}[\text{TPR}|\mathtt{m}]| + |\mathbf{E}[\text{FPR}|\mathtt{m}] - \mathbf{E}[\text{TPR}|\mathtt{f}]| - \epsilon$.

**R1: How is the candidate selection done? What is the algorithm to optimize over different candidate sets?** In
our experiments, RobinHood used CMA-ES [45] to find candidate solutions. RobinHood randomly partitions the data
into 60% candidate and 40% safety data sets. One avenue of future work we will discuss is to optimize this partitioning
to maximize the probability of success. **R1: How does Algorithm 1 update the iteration, or how to construct the**
**set $\Theta$? Is it agnostic to the ML used?** Algorithm 1 relies on the feasible set and optimization algorithm (OA) the user
chooses to find candidate solutions. It is agnostic w.r.t. that OA. However, if the user chooses a poor OA that cannot
find solutions, our approach will return NSF. We found that CMA-ES [45] works well. We will make it clear that the
ability of our algorithm to find a fair solution depends on the user's choice of OA.

**R1: The word 'fair' is problematic. Whether a solution is fair depends on more than the statistics of the model.**
Thank you for pointing out our imprecise wording. We will clearly differentiate between ensuring that models are fair,
and ensuring that fairness constraints defined by the user are satisfied. We do the latter, and will not claim the former. It
is the user's responsibility to provide a $g(\theta)$ that captures their notion of fairness for the application at hand; if the user's
definition does not sufficiently capture fairness, then the solutions RobinHood produces will not either. RobinHood is
designed to give the user flexibility in providing fairness definitions that capture domain knowledge.

**R1: In line 261, fairness is mentioned without a proper explanation of why that is fair.** We will make our writing
more clear and rigorous in what we mean by fair in this experiment. Our definition is just one choice; RobinHood can
be applied with other definitions of fairness the user finds more relevant.

**R1: Limitations regarding the ability to "satisfy multiple criteria."** Thank you for pointing this out. We will clarify
that our results do not contradict the references you provide, and discuss the theoretical limitations. RobinHood returns
NSF when impossibilities such as conflicting fairness constraints exist.

**R2: Col 3 in figs suggests that baseline algs approach the same failure rate as RobinHood given enough data.**
**Any insights?** The fairness-unaware baselines only try to maximize expected reward. When reward maximization and
fairness are nonconflicting, there can exist fair high-performing solutions. When *only* the high-performing solutions
are fair, the failure rate of reward maximization algorithms should decrease as more data is provided. Note that when
reward maximization and fairness *are* conflicting, e.g., in the skewed proportions experiment, the failure rates of the
unfair baselines do not diminish. Importantly, while the baselines might be fair in some cases, unlike RobinHood, these
approaches do not provide fairness guarantees.

**R3: Regarding "no mention of the support assumption".** This is captured by Assumption 4 for Thm 2, but is
something we should, and will, discuss around (2) in the supplemental. **R3: Wouldn't it be ideal to return a uniform**
**random policy as the solution rather than an NSF?** If no fair policy exists, RobinHood returns NSF. The user has
control over what to do in this case. For some domains, deploying a known fair policy, (e.g., uniform random) may be
appropriate; for others, it might be more appropriate to issue a warning and deploy no policy.

**R3: Why is inflateBounds needed to compute the candidate utility but not when certifying fairness?** The candi-
date selection method (CSM) searches for a solution that will pass the safety test (ST), which requires testing multiple
solutions. Essentially, the CSM is performing multiple comparisons with one data set, resulting in over-estimation of its
confidence that the solution it picks will pass the ST. This results in RobinHood frequently returning NSF. Inflating the
width of the confidence intervals in the CSM is an effective remedy. Note that this multiple comparisons problem does
not impact the ST, which only tests one solution, and so does not invalidate our theoretical guarantees.

[Meta-Review · NeurIPS 2019]

The reviewers reached a clear consensus on this paper after careful reviews, reading through the author response, and participating in a discussion about this paper. The reviewers found that this paper makes an important contribution to the field in enforcing fairness constraints for offline contextual bandits, and is well supported theoretically and a solid empirical evaluation for confirmation. The reviewers did find several areas where the paper could be strengthened, most especially in moving as much of the formal analysis, proofs, or at least proof sketches into the main text. The authors have responded well to these suggestions and I expect that the final version of this paper will incorporate these changes to the full extent possible within the space constraints. Overall, this seems to be a strong paper and well worth the attention of the NeurIPS community.